# Not so griddy: Internal representations of RNNs path integrating more than one agent

**William T. Redman**[*]
Intelligent Systems Center
Johns Hopkins Applied Physics Laboratory

**Francisco Acosta**
Department of Physics
University of California, Santa Barbara

**Santiago Acosta−Mendoza**
Dynamical Neuroscience
University of California, Santa Barbara

**Nina Miolane**
Department of Electrical & Computer Engineering
University of California, Santa Barbara

## Abstract

Success in collaborative and competitive environments, where agents must work with or against each other, requires individuals to encode the position and trajectory of themselves and others. Decades of neurophysiological experiments have shed light on how brain regions [e.g., medial entorhinal cortex (MEC), hippocampus] encode the self's position and trajectory. However, it has only recently been discovered that MEC and hippocampus are modulated by the positions and trajectories of others. To understand how encoding spatial information of multiple agents shapes neural representations, we train a recurrent neural network (RNN) model that captures properties of MEC to path integrate trajectories of two agents simultaneously navigating the same environment. We find significant differences between these RNNs and those trained to path integrate only a single agent. At the individual unit level, RNNs trained to path integrate more than one agent develop weaker grid responses, stronger border responses, and tuning for the *relative* position of the two agents. At the population level, they develop more distributed and robust representations, with changes in network dynamics and manifold topology. Our results provide testable predictions and open new directions with which to study the neural computations supporting spatial navigation.

## 1 Introduction

When navigating real-world environments, individual agents (e.g., animals) must adjust their trajectories based on the observed trajectories of others [1, 2]. This becomes especially important in competitive and collaborative settings, where the positions, movement directions, and speeds of others contain information necessary for goal-based decision making. For example, an agent competing with others for food must keep track of where "opponent" agents are and where they are headed, in order to effectively update its strategy. While decades of research has shed fundamental light on the neural circuitry involved in spatial navigation in the absence of others ("single agent" navigation), little is known about how the brain enables spatial navigation in the more general "multi-agent" scenario.

A core component of spatial navigation is *path integration* [3], the ability to update an estimate of position in an environment from a sequence of movement directions and speeds (Fig. 1A). Medial entorhinal cortex (MEC) is believed to play a critical role in supporting path integration via functional classes of neurons [4], such as grid cells [5, 6], border cells [7], and band cells [8]. Recent research has shown that MEC is activated when human subjects observe others navigate

---

[*]will.redman@jhuapl.edu

38th Conference on Neural Information Processing Systems (NeurIPS 2024).

environments [9, 10], suggesting that it may additionally be involved in computations supporting multi-agent path integration. That neural activity in MEC could be shaped by trajectories of others is further supported by recent work demonstrating that populations of cells in hippocampus and anterior cingulate cortex, brain regions interconnected with MEC, encode spatial information related to the positions of other agents [11, 12, 13, 14, 15, 16, 17]. However, what representations exist in MEC to support multi-agent path integration and how they differ from the representations that have been found in single agent environments, remains largely unexplored. This is of additional importance as it is unclear whether continuous attractor networks (CANs) [18, 19, 20], which are believed to underlie the grid code in MEC [21, 22, 23, 24, 25], can simultaneously support the encoding of multiple individuals [26, 27].

To begin to investigate this open research direction, we train a recurrent neural network (RNN) model to perform path integration of two agents ("dual agent" path integration[2]) and interrogate the representations that emerge. These RNN models [28, 29], when trained on single agent path integration, have been shown to develop properties analogous to MEC. In particular, they exhibit units with grid, border, and band responses [28, 29, 30] and population dynamics consistent with continuous toroidal attractors [31, 32].

While their biological plausibility remains unclear [33, 34], we choose to analyze these RNN models for four reasons. First, they contain inter-connectivity between "hippocampus"−like and "MEC"−like layers, enabling us to model aspects of the experimentally reported multi-agent hippocampal responses [11, 12, 13, 14, 16, 17]. Second, the lack of neural data from MEC in multi-agent settings makes the outputs of these RNN models unbiased predictions that can be used to further assess their validity. Third, these RNN models have been shown to have response properties beyond grid, border, and band cells that are similar to those in MEC recordings [35], suggesting that the RNNs learn solutions more broadly representative of MEC. And fourth, beyond their connection to neuroscience, these RNN models represent legitimate machine learning systems that exhibit improved goal-based navigation when paired with reinforcement learning [28]. As autonomous agents becoming increasingly deployed in everyday environments, their ability to navigate in multi-agent settings becomes increasingly important [36] and understanding how the representations learned by RNN models trained on single and dual agent path integration compare can provide new insight into how to improve this capability.

We find that, with only a few modifications, we can train existing RNN models [28, 29] to achieve good performance on dual agent path integration ("dual agent RNNs"−Sec. 2). These dual agent RNNs are flexible enough to generalize to single agent path integration, whereas RNNs trained on single agent path integration ("single agent RNNs") are not capable of performing dual agent path integration (Sec. 3.1), demonstrating how representations that are optimal in the single agent setting need not be in the dual agent setting. Investigating the learned representations, at the individual unit level, we find that the need to path integrate more than one agent leads to the emergence of stronger border representations and weaker grid representations (Sec. 3.2), a result consistent with recent human neurophysiological experiments [9, 10]. Measuring the representations in a reference frame that captures the joint position of both agents, we find that units in dual agent RNNs develop tuning for "relative space" (Sec. 3.3). Collectively, these results provide insight into how the dual agent RNN is able to efficiently solve its task. Utilizing topological and dynamical tools, we characterize differences between single and dual agent RNNs at the population level (Sec. 3.4), finding no evidence for continuous toroidal attractors underlying the activity of dual agent RNNs which further emphasizes their distinct computational mechanisms. These results question how our existing understanding of MEC can be extended to the ethologically relevant multi-agent setting, and we hope our that they will motivate future experimental and theoretical exploration.

**Contributions.** We study, *for the first time*, how representations that emerge in RNNs trained on dual agent path integration differ, at the individual unit and population level, from the representations of single agent RNNs. Our results are consistent with recent neurophysiological experiments [9, 10] and provide testable predictions that can guide future experimental design.

---

[2]While there is no direct evidence for simultaneous dual agent path integration, we hypothesize that the brain can perform it and that the MEC is involved. See Appendix A for our rationale.

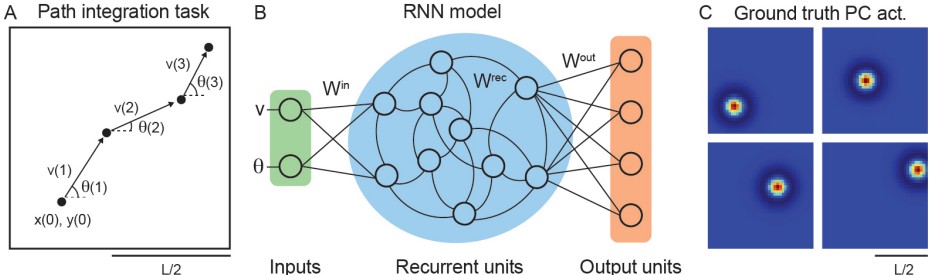

Figure 1: **Overview of RNNs trained to perform single agent path integration.** (A) Illustration of path integration task. The agent starts at known position $x(0)$ and $y(0)$ and makes a sequence of movements through space given by the movement directions $\theta(t)$ and speeds $v(t)$. From this, $x(t+1)$ and $y(t+1)$ must be estimated. (B) Schematic of single agent RNN model architecture [28, 29], which takes input $v(t)$ and $\theta(t)$, recurrently processes it, and drives activations of an output layer. (C) Example ground truth place cell (PC) activations used as targets for the RNN output units.

## 2  RNN model

Prior work has considered the setting where a single agent traverses a square environment through finite length paths [28, 29, 30]. The agent starts a given path at position $\mathbf{z}(0) = [x(0), y(0)]^{\mathsf{T}} \in [-\mathrm{L}/2, \mathrm{L}/2] \times [-\mathrm{L}/2, \mathrm{L}/2]$ and subsequently moves to positions $\mathbf{z}(0) \to \mathbf{z}(1) \to ... \to \mathbf{z}(T)$. This path can alternatively be defined by $\mathbf{z}(0)$ and a sequence of movement directions, $\theta(t) \in [-\pi, \pi]$, and speeds, $v(t) \in [0, \infty)$, such that $\mathbf{z}(t+1) = \mathbf{z}(t) + v(t)\Delta t \cdot [\cos\theta(t), \sin\theta(t)]^{\mathsf{T}}$, for small $\Delta t \in \mathbb{R}^+$ (Fig. 1A). The accurate updating of $\mathbf{z}(t+1)$ from only $\mathbf{u}(t) = [\theta(t), v(t)]^{\mathsf{T}}$ is formally what is meant by **path integration**.

To learn an RNN model that can perform single agent path integration, prior work has utilized the following architecture (Fig. 1B). The inputs $\mathbf{u}(t)$ are projected, via weights $\mathbf{W}^{\text{in}} \in \mathbb{R}^{n_G \times 2}$, into a recurrent layer of $n_G$ units. This recurrent layer has connectivity defined by weights $\mathbf{W}^{\text{rec}} \in \mathbb{R}^{n_G \times n_G}$. The units in the recurrent layer are used to drive the activation of $n_P$ output units, via weights $\mathbf{W}^{\text{out}} \in \mathbb{R}^{n_P \times n_G}$. More explicitly, the RNN model evolves via the dynamics

$$\mathbf{r}(t+1) = \sigma\left[\mathbf{W}^{\text{rec}}\mathbf{r}(t) + \mathbf{W}^{\text{in}}\mathbf{u}(t)\right], \tag{1}$$

$$\hat{\mathbf{p}}(t+1) = \mathbf{W}^{\text{out}}\mathbf{r}(t+1), \tag{2}$$

where $\mathbf{r}(t) \in \mathbb{R}^{n_G}$ and $\hat{\mathbf{p}}(t) \in \mathbb{R}^{n_P}$ are the activations of the recurrent and output units at time $t$, and $\sigma(\cdot)$ is an activation function.

The RNN model, defined by Eqs. 1−2, is trained by comparing the output layer activations, $\hat{\mathbf{p}}(t)$, with "ground truth" activations, $\mathbf{p}(t)$, using the cross entropy loss,

$$\mathcal{L}(t) = -\hat{\mathbf{p}}(t)\log\mathbf{p}(t) - \lambda||\mathbf{W}^{\text{rec}}||_2. \tag{3}$$

The second term, $\lambda||\mathbf{W}^{\text{rec}}||_2$, is a weight decay regularization term that penalizes the $L_2$ norm of the recurrent weights, by strength $\lambda \in \mathbb{R}^+$. This was found to be beneficial for generalization [32]. The form $\mathbf{p}(t)$ takes has been shown to determine properties of the representations learned by the single agent RNNs [29]. We follow Sorscher et al. (2019) and set the ground truth activations to be "place cell"−like [37], with the place fields having a difference of Gaussian (DoG) structure [29] (Fig. 1C),

$$p_i(t) = \text{DoG}[\mathbf{z}(t), \mathbf{c}_i] = \exp\left(-||\mathbf{z}(t) - \mathbf{c}_i||_2^2/2\sigma_1^2\right) - \exp\left(-||\mathbf{z}(t) - \mathbf{c}_i||_2^2/2\sigma_2^2\right), \tag{4}$$

where $p_i(t)$ is the ground truth activation of unit $i$ at time $t$, $\mathbf{c}_i \in [-\mathrm{L}/2, \mathrm{L}/2] \times [-\mathrm{L}/2, \mathrm{L}/2]$ is the center of its place field, and $\sigma_1 < \sigma_2$ define the place field's size. The initial activation in the recurrent layer is set using the ground truth place cell activations, $\mathbf{r}(0) = \mathbf{W}^{\text{back}}\mathbf{p}(0)$, where $\mathbf{W}^{\text{back}} \in \mathbb{R}^{n_G \times n_P}$ is a learnable set of weights.

To assess the RNN model's performance in single agent path integration, the position of the agent is decoded from the activations of the output layer. Prior work has used a simple top-$n_d$ decoder, where the predicted location of the agent is found by averaging the place field centers, $\mathbf{c}_i$, corresponding

to the $n_d \in \mathbb{N}$ units in the output layer with the highest activations. That is, $\hat{\mathbf{z}}(t) = \frac{1}{n_d} \sum_{i=1}^{n_d} \mathbf{c}_{\alpha_i}(t)$, where the $\alpha_i$ correspond to the ranked order of the output layer activations [i.e., $\hat{p}_{\alpha_1}(t) > \hat{p}_{\alpha_2}(t) > ... > \hat{p}_{\alpha_{n_P}}(t)$]. The decoding error is then defined as $||\mathbf{z}(t) - \hat{\mathbf{z}}(t)||_2$.

To enable the RNN model to perform dual agent path integration, we make four modifications (see Appendix B and Table S1, for more details):

- **Additional inputs:** We set $\mathbf{u}(t) = [\theta_1(t), v_1(t), \theta_2(t), v_2]^\mathsf{T}$, where $\theta_i(t)$ and $v_i(t)$ correspond to the $i^{\text{th}}$ agent's movement direction and speed at time $t$.

- **Summed place cell activations:** We set the ground truth place cell responses to be the sum of their responses to the location of each agent independently. That is, if $\mathbf{z}_i(t)$ corresponds to the position of the $i^{\text{th}}$ agent at time $t$, then $p_i(t) = \text{DoG}[\mathbf{z}_1(t), \mathbf{c}_i] + \text{DoG}[\mathbf{z}_2(t), \mathbf{c}_i]$. This choice, inspired by "social" place cells [11, 12, 13, 14, 16, 17], is an important one and we provide our rationale in choosing it in Appendix B.1.

- $k$-**means clustering for decoding:** We use $k$-means clustering to separate the place field centers corresponding to the $2n_d$ output layer units with the highest activations into $k = 2$ groups. The predicted positions of the two agents, $\hat{\mathbf{z}}_1(t)$ and $\hat{\mathbf{z}}_2(t)$, are estimated by the center of each cluster.

- **Reduced weight decay regularization:** We reduce the weight regularization strength, $\lambda$, in the loss function (Eq. 3), as we expect the added complexity of dual agent path integration to require more of $\mathbf{W}^{\text{rec}}$'s capacity.

Training this modified RNN on dual agent path integration, we show−for the *first time*−that it is possible to learn a network that achieves high performance (Figs. 2, S1). While the decoding error of dual agent path integration is greater for dual agent RNNs than the decoding error of single agent path integration is for single agent RNNs (Fig. 2A, B), we find that nearly $50\%$ of all 5000 dual agent trajectories sampled have an error below 0.10 m, which was previously used as a threshold to determine success [33] (Fig. 2B). In addition, visualizing the RNN's prediction on individual dual agent trajectories shows qualitatively similar behavior to the true trajectories, even when decoding error is $> 0.10$ m. (Fig. 2C). Examining the trajectories where the two agents are closest to each other (Fig. S2), we find the dual agent RNN is able to maintain its performance. As hypothesized, reducing the strength of weight decay, from $\lambda = 10^{-4}$ to $\lambda = 10^{-6}$, enables an increase in performance on dual agent path integration (Fig. 2A−compare pink solid and dashed lines).

## 3 Results

### 3.1 Representations in single agent RNNs are not optimal for dual agent path integration

The discovery that recurrent units in RNNs trained on single agent path integration (Sec. 2) develop responses like those found in MEC [28, 29, 30, 35] has been interpreted as a normative demonstration of the optimality of these representations. Are these same representations also optimal for dual agent path integration?

To test this, we apply trained single agent RNNs to the dual agent setting (Appendix C). These networks have a decoding error of $\approx 1.0$ m. (Fig. 3A−red dot), nearly a $10\times$ increase in error relative to the trained dual agent RNNs (Fig. 3A−pink solid line). To determine whether this high decoding error is due to poor generalization or to the RNN being sensitive to the precise value of the network weights, we "fine-tune" all the trained single agent RNN weights ($\mathbf{W}^{\text{rec}}, \mathbf{W}^{\text{in}}, \mathbf{W}^{\text{out}}, \mathbf{W}^{\text{back}}$) for 10 epochs on the dual agent path integration task (Appendix C). We find that this leads to an improvement in performance on dual agent path integration, although the end decoding error remains higher than the decoding error of the trained dual agent RNN (Fig. 3A−compare solid red and pink solid lines). However, this improvement is dependent on updating the trained weights in the recurrent layer, as freezing $\mathbf{W}^{\text{rec}}$ and fine-tuning the other weights again leads to high decoding error (Fig. 3A−red dashed line). Taken together, these results demonstrate that the representations learned by the recurrent units in trained single agent RNNs are not optimal for dual agent path integration.

In contrast, trained dual agent RNNs can perform single agent path integration with decoding error $< 0.10$ m. (Fig. 3B−green dot). Fine-tuning the weights on single agent path integration further improves performance, even when $\mathbf{W}^{\text{rec}}$ is frozen (Fig. 3B−green dashed line). This is aligned with

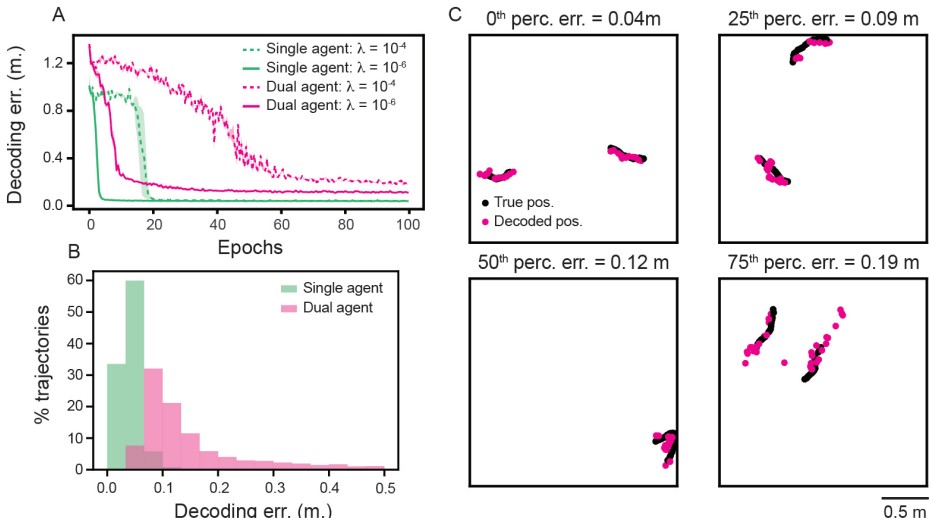

Figure 2: **RNNs can be trained to successfully perform dual agent path integration.** (A) Decoding error, as a function of training epoch, for RNNs trained and tested on single and dual agent path integration. Solid line is mean across 5 independently trained networks and shaded area is maximum and minimum of all 5 networks. (B) Distribution of median decoding error, across 5000 trajectories (1000 per network), for single and dual agent RNNs. (C) Example ground truth and decoded dual agent trajectories. Trajectories were chosen as those closest to the $0^{th}$, $25^{th}$, $50^{th}$, and $75^{th}$ percentile of the decoding error distribution.

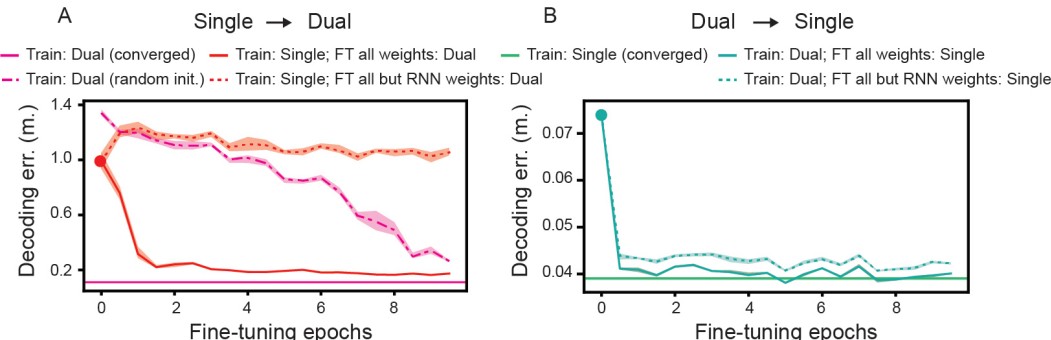

Figure 3: **Dual agent RNNs can generalize to single agent path integration, but not vice versa.** (A) Decoding error of RNNs trained on single agent path integration and tested on dual agent path integration with (red lines) and without (red dot) fine-tuning (FT) on dual agent path integration. Converged performance and performance from random initialization of dual agent RNNs (dashed and solid pink lines, respectively) are shown for comparison. (B) Same as (A), but for RNNs trained on dual agent path integration and tested on single agent path integration. (A)−(B) Lines denote mean and shaded area denotes maximum and minimum, across 5 independently trained RNNs.

recent experimental and computational work showing that fixed MEC connectivity and slow time-scale plasticity between MEC and hippocampus enables generalizable representations of new environments [38]. These results (which exhibit a similar trend when comparing the training loss instead of the decoding error−Fig. S3) demonstrate that dual agent RNNs have internal representations that are flexible enough to enable generalization to single agent path integration.

## 3.2 Single and dual agent RNNs develop different representations at the individual unit level

To begin dissecting the representations that emerge in dual agent RNNs, we examine the functional properties of individual units in the recurrent layer. Previous work has found that single agent RNNs develop responses analogous to three functional classes of neurons in MEC: grid cells [5, 6], border cells [7], and band cells [8]. The distribution of these properties can be quantified by computing scores

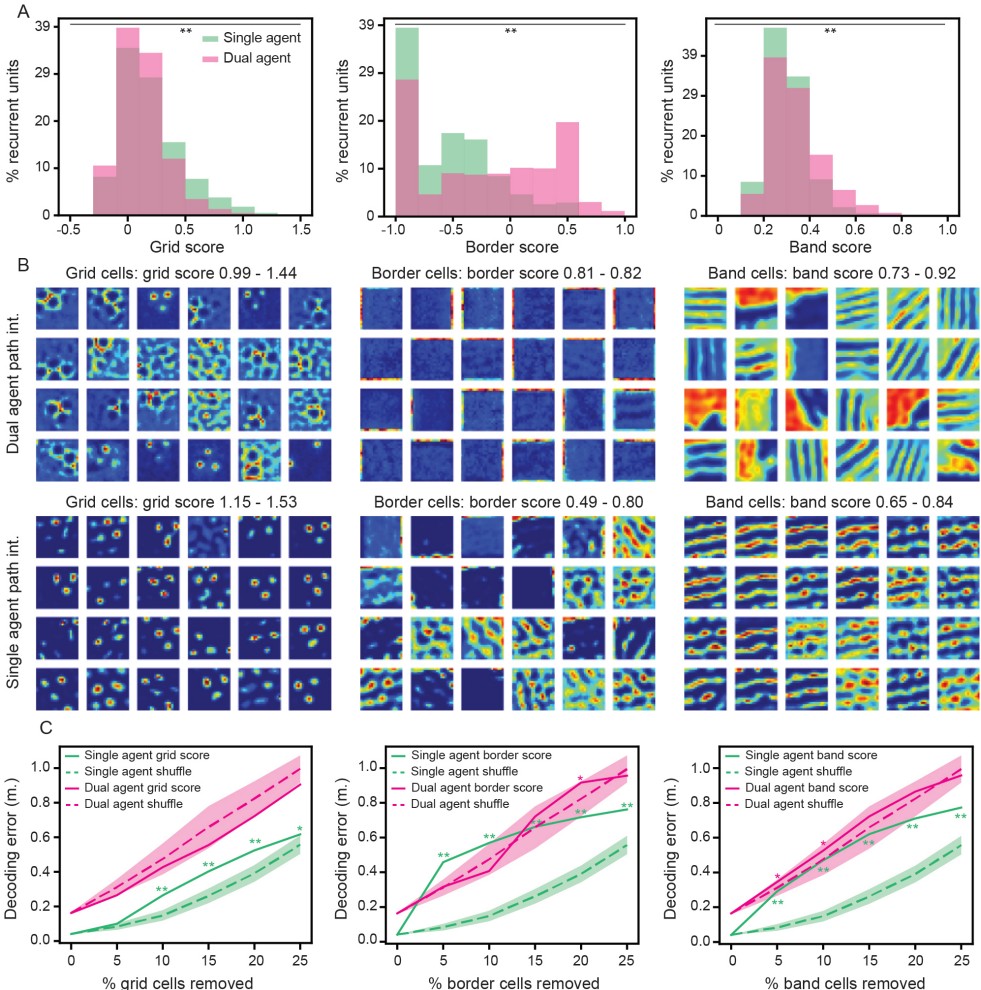

Figure 4: **Single and dual agent RNNs differ in their distribution of functional properties.** (A) Distribution of grid, border, and band scores, computed for all units of single and dual agent RNNs. Distribution includes 5 independently trained RNNs. Kolmogorov-Smirnov (KS) test used to compare distributions ($** : p < 0.01$). (B) Visualization of rate maps (Appendix D.1) for units, from individual RNNs, with highest grid, border, and band scores. (C) Decoding error of trained RNNs with the units corresponding to the highest grid, border, and band score ablated (solid line: mean across 5 independently trained RNNs), compared to trained RNNs with random units ablated (dashed lines: mean across 10 randomly sampled choices of ablated units for each independently trained RNN; shaded area: $\pm$ standard deviation) (Appendix D.3). Mann-Whitney test is used to compare distributions ($* : p < 0.05$ and $** : p < 0.01$—see Table S2 for $p$-values).

for each unit (Appendix D.2). While grid and border scores are standard metrics, to our knowledge no band score exists. We therefore developed a metric to quantify "banded−ness" (Appendix D.2).

We find statistically significant differences in the distribution of these functional properties between single and dual agent RNNs (Fig. 4A, B). In particular, we find dual agent RNNs develop units with lower grid scores, but higher border and band scores (Fig. 4A). Visualizing rate maps associated with the dual agent RNN units that have the highest grid scores, we find that many have responses that are not solely constrained to vertices of triangular lattices, but are instead "honeycomb"−like (Fig. 4B). This suggests that dual agent path integration leads to a weaker grid code. In contrast, dual agent RNN units with high border and band scores have more "ideal" responses, as compared to units of single agent RNNs, which have additional grid structure (Fig. 4B). This demonstrates that dual agent path integration leads to a strengthened border and band code. These results are consistently observed across all 5 independently trained RNNs (Fig. S4).

Prior work has found all three of these classes to be important for single agent path integration [28, 30, 31, 32, 39, 40]. To understand their importance in dual agent path integration, we perform targeted ablations, removing units with the highest grid, border, and band scores (Appendix D.3). We compare the decoding error of these RNNs to RNNs with randomly removed units, as this has been found to be a strong baseline [31, 35]. Unlike single agent RNNs, which have a significant increase in decoding error relative to the random baseline with the ablation of all functional classes, dual agent RNNs are less sensitive to the ablation of any one functional class (Fig. 4C). This suggests that dual agent RNNs develop a more distributed and robust representation. We note that the greater (albeit, small) sensitivity of the dual agent RNNs to ablations of border and band cells are consistent with work arguing their driving role in path integration [30, 31, 39, 40].

To determine the extent to which our results are due to the nature of dual agent path integration, as opposed to specific choices of RNN architecture and hyper-parameters, we perform several control experiments. First, we consider the possibility that, by having twice as many agents to track, dual agent RNNs have half the effective size of single agent RNNs. In such a case, we might expect a smaller single agent RNN to generate responses similar to those of dual agent RNNs. However, half-sized single agent RNNs have representations that are distinct from the full-sized dual agent RNNs (Fig. S5A). Second, dual agent rate maps are computed using the position of only one of the two agents (the other agent being "averaged out"−Appendix D.1). We re-compute the rate maps of dual agent RNNs, removing one of the agents completely. Thus, while the network was trained in the dual agent setting, the rate maps are only affected by a single agent. We find this to have little effect on the representations (Fig. S5B). Third, we vary the agent whose position is being used to construct the rate map. We find that the rate maps are nearly identical (Fig. S6). Given that there is nothing in the loss function (Eq. 3) that distinguishes the two agents, it is not surprising that the dual agent RNN treats them the same. Finally, we measure the distribution of grid, border, and band scores for the RNNs trained with greater weight decay regularization (Fig. 2A−dashed lines). We find that the overall trend of dual agent path integration leading to lower grid scores and higher border scores, relative to single agent path integration, holds (Fig. S7).

Taken together, these results demonstrate that the addition of another dynamic variable (i.e., a second agent) leads to differences in the distribution of functional properties, at the individual unit level, between single and dual agent RNNs.

## 3.3  Dual agent RNNs develop tuning in relative space

Recent neural recordings have demonstrated that, when mice are trained to navigate to reward sites that move in physical space, grid cells perform path integration in reference frames that are anchored to the locations of these rewards [41]. These results highlight the fact that hippocampus and MEC can perform computations in different coordinate systems [42]. Inspired by these results, we examine whether dual agent RNNs develop grid responses in another reference frame. One candidate coordinate system is defined by the relative positions of the two agents ("relative space"−Fig. 5A, Appendix E). We do not find evidence for units having grid responses in relative space (Fig. 5B, C), demonstrating that dual agent RNNs are not simply using the same representations as in the single agent setting, but in a different reference frame. However, we find that units classified as border and band cells in the absolute (i.e., allocentric) reference frame have strong tuning in relative space (Fig. S8A). Units with high border score have greatest activations when both agents are near borders (Fig. S8A, B) and units with high band score have greatest activations when both agents are positioned at periodic distances, along parallel lines (Fig. S8A, B). The latter could enable easy representation of parallel movement, where the agents are naturally separated. In contrast, units with high grid score have activations that are largely invariant to relative space (Fig. S8A, B).

To further characterize the extent to which dual agent RNNs encode relative position, we compute the spatial information [43] of all recurrent units relative space rate maps (Appendix E.2). We find that, while most units contain little spatial information in relative space (corresponding to spatial information $\approx 0$), the distribution is heavy tailed (Fig. 5D). Visualizing the relative space rate maps with the highest spatial information, we find that many are localized, with greatest activation when the two agents are near each other (Fig. 5E−center pixels). We hypothesize that, because accurate dual agent path integration becomes more challenging as the two agents get closer together, ablating units with high spatial information in the relative space reference frame would lead to an increase in decoding error. In-line with this hypothesis, we find that removing the units with the highest

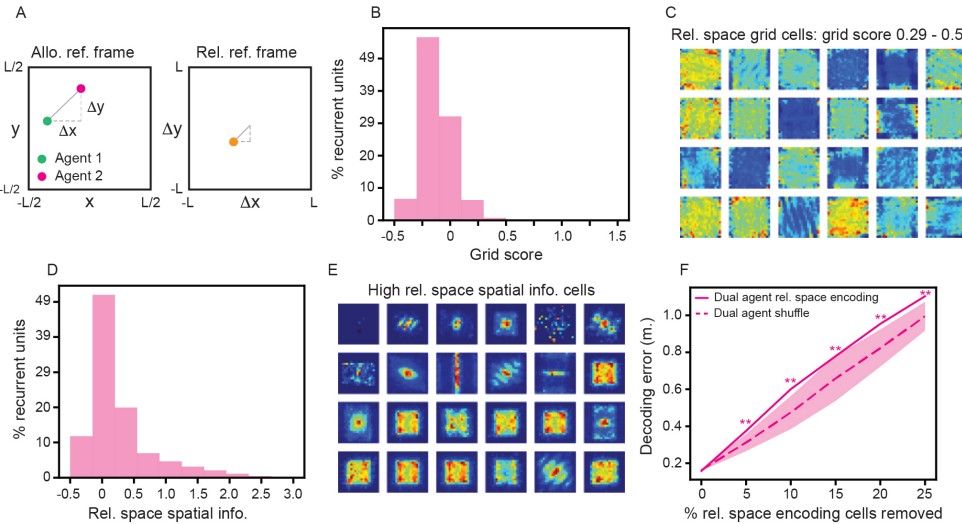

Figure 5: **Dual agent RNNs develop tuning to "relative space".** (A) Schematic illustration of a transformation from the allocentric reference frame to the relative space reference frame. (B) Distribution of grid scores (across 5 independently trained RNNs), computed on the relative space rate maps (Appendix E.1). (C) Visualization of relative space rate maps, from an individual dual agent RNN, with the highest relative space grid scores. (D) Distribution of spatial information, computed on the relative space rate maps. Distribution includes 5 independently trained RNNs. (E) Visualization of relative space rate maps, from an individual dual agent RNN, with the highest relative space spatial information. (F) Decoding error of dual agent RNNs with units having the highest relative space spatial information ablated. Dashed line and shaded area same as Fig. 4C. Mann-Whitney test is used to compute $p$-values ($** : p < 0.01$—see Table S3 for statistics).

relative space spatial information leads to a statistically significant decrease in decoding performance, relative to the random ablation baseline (Fig. 5F, Appendix E.3). That this ablation leads to stronger deficits than the ablation of grid, border, and band cells (Fig. 4C), emphasizes of their importance in supporting dual agent path integration. Further, it demonstrates that the dual agent RNN utilizes an efficient representation (encoding the *relative* position) to achieve dual agent path integration.

### 3.4 Single and dual agent RNNs learn different representations at the population level

Our comparison of the representations that emerge in single and dual agent RNNs has thus far been focused on the activations of individual recurrent units. Our findings that dual agent RNNs develop different distributions of functional classes than single agent RNNs, and that dual agent RNNs encode relative space spatial information, are suggestive that the network structure underlying single and dual agent RNNs are distinct. To further explore this hypothesis, we analyze the recurrent unit activations at the population level.

Topological data analysis (TDA) [44] has become a widely used tool for studying neural population activity [25, 45, 46, 47, 48, 49, 50]. A core technique in TDA is persistent homology (Appendix F.1), which can be used to estimate the topology of the underlying neural manifold. Applying persistent homology, we find evidence that the population activations of single agent RNNs are constrained to a manifold having topology consistent with a two-dimensional torus (Fig. 6A, left−black arrows), consistent with prior characterizations [31, 32]. When performing the same analysis on the population activations of dual agent RNNs, we find differences in the persistent homology. In particular, the topological signatures of a two-dimensional torus are not clearly present in the persistence diagram (Fig. 6A, right). We see similar patterns in the persistence diagrams across different independently trained single and dual agent RNNs (Fig. S9). This difference in topology can further be seen by the fact that the persistence diagrams corresponding to single agent RNNs are closer (with respect to an appropriate metric [51]) to the persistence diagrams of an idealized torus, than the persistence diagrams corresponding to the dual agent RNN are (Fig. S10). This suggests that dual agent RNNs **do not** have the same continuous attractor structure that single agent RNNs have [31, 32].

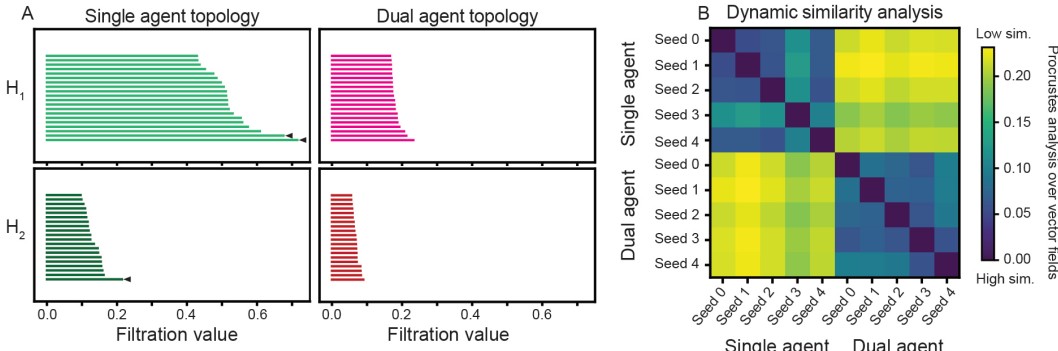

Figure 6: **Population level activations of single and dual agent RNNs differ in their topology and dynamics.** (A) Persistence diagrams (Appendix F.1) of the population activations of example single and dual agent RNNs. $H_1$ bars of high filtration value correspond to existence of loops and $H_2$ bars correspond to existence of two-dimensional cavities. Black arrows denote features of the single agent persistence diagram that are consistent with a two-dimensional toroidal attractor. (B) DSA (Appendix F.2) applied to the activations of independently trained single and dual agent RNNs. Color denotes Procrustes analysis over vector fields metric.

The topology and geometry of neural manifolds do not capture all properties of the underlying neural circuit. Indeed, RNNs with the same dynamical properties (e.g., existence of fixed point attractors, limit cycles) can be constructed to have different population activation geometries [52]. Similarly, RNNs with the same population activation geometries can be constructed to have different dynamics [52]. Recent work has developed a data-driven dynamical systems framework [dynamic similarity analysis (DSA)] for computing compact representations of RNN dynamics [53]. Therefore, we complemented our TDA results by applying DSA (Appendix F.2) to the population activations of single and dual agent RNNs. We find that independently trained single agent RNNs are dynamically more similar to each other than they are to independently trained dual agent RNNs, and vice versa (Fig. 6B). Therefore, at both the topological and dynamical level, the representations learned by the single and dual agent RNN are different, suggesting that the addition of a second agent fundamentally changes the network structure that emerges.

## 4 Discussion

Inspired by recent work demonstrating that brain regions believed to encode the self's position and trajectory (e.g., MEC, hippocampus) are modulated by the position and trajectories of others [9, 10, 11, 12, 13, 14, 16], we trained a class of RNN models [28, 29] to path integrate two agents and investigated the representations that emerged. Given that these RNN models, when trained to path integrate single agents, develop properties that are similar to those found in MEC at the individual unit (e.g., grid, border, and band responses) and population (e.g., network structure with signatures of continuous toroidal attractors) level, this is a natural point for starting to address the larger question of how the MEC supports representation of multiple dynamic variables.

We find that the representations that emerge in dual agent RNNs significantly differ from those of single agent RNNs. Dual agent path integration leads to weaker grid codes and stronger border codes (Fig. 4). This is consistent with electrophysiological recordings that have found MEC encodes distance of others to boundaries [9] and fMRI recordings that have observed a negative correlation between grid cell strength and performance on multi-agent tasks [10]. Functional class ablations and TDA revealed a distributed and robust representation, with no evidence for the population activity being constrained to a two-dimensional torus (Fig. 6). Despite this, dual agent RNNs are able to generalize to perform single agent path integration, even when the weights in their recurrent layer are frozen (Fig. 3B). This demonstrates the flexibility of their representations [38] and is in-line with work finding that neural networks trained to perform multiple tasks develop abstract and distributed tuning [54, 55, 56, 57]. Finally, we found that RNNs trained on dual agent path integration develop tuning in the coordinate system defined by the relative position of the two agents (Figs. 5, S8). This is consistent with the tuning of neurons in anterior cingulate cortex to relative position between agents [15], as well as is aligned with neurophysiological experiments that have uncovered neurons in MEC that encode distance and orientation of self to objects [58].

**Limitations.** The biological plausibility of the RNN models investigated in this work [28, 29] have been questioned [33, 34] (although, see response by Sorscher et al. (2022) [59]). While our dual agent ground truth place cells were motivated by experimental work measuring hippocampal responses in environments with multiple animals [11, 12, 13, 14, 16, 17], it represents only one possible choice (as discussed in Appendix B). Additionally, it has been found that the RNN model develops weaker modularization in grid properties [28, 29, 30], a feature that is believed to be critical for encoding local spatial information [60, 61, 62] (although see work suggesting individual grid modules contain more spatial information than previously thought [63, 64, 65, 66]). Recent use of loss functions that incorporate more than just path integration have found stronger grid responses and modularization [67, 68]. Future work should investigate the properties that emerge in these models when tasked with dual agent path integration. In our experiments, we did not account for noise in velocity inputs, which have been found to play a dominant role in human path integration error [69], and are presumably higher for the assessment of others' velocities (given the lack of vestibular and proprioceptive feedback). We hypothesize that, as noise is added to the estimate of one agent's position and velocity, dual agent RNNs will develop representations more similar to single agent RNNs, suggesting a trade-off that should be explored.

**Predictions.** Recent advances in experimental paradigms, where rodents must update their goal-based planning given the trajectories of others [70, 71, 72], offer the possibility of comparing the *in vivo* representations of MEC with those generated by the dual agent RNN. Our results make several testable predictions about what responses might be found in MEC. First, we expect grid responses to be weakened, with the cells that do have high grid score exhibiting deviations from standard grid cells ("honeycomb"−like responses−Fig. 4). Second, we expect strengthened boundary and border responses (Fig. 4), with border cells responding most strongly when both agents are near borders (Fig. S8). Finally, we expect that the relative position between agents will be encoded in the a subpopulation of MEC neurons (Fig. 5) that is disjoint from cells with high grid scores (Fig. S8). Testing these predictions will provide insight into the validity of the RNN model, as well as our assumption that MEC can perform simultaneous path integration of two agents.

**Future directions.** There are several impactful directions that our work can be extended. First, analyzing the emergent properties of RNNs trained to path integrate more than two agents can lead to a better understanding of how the MEC performs computations in the multi-agent setting. Second, reinforcement learning can be used to probe how the dual agent RNN representations might support higher order behavior [73, 74, 75] (such as shortcut taking in dynamic chasing [76]). Third, theoretical work, showing−from first principles−what representations are necessary for single agent path integration [77, 78] can be extended to the dual agent path integration setting. This will provide a thorough treatment of how the complexity of multi-agent environments changes the computations needed to be performed by MEC. Finally, foundational work on MEC has found that grid cells encode abstract relationships beyond purely spatial variables [79]. These include social hierarchies [80], sound frequencies [81], list ordering distances [82], and elapsed time [83, 84, 85]. Our work has implications for how the MEC may simultaneously integrate information across these different cognitive axes.

**Outlook.** Continuous attractor networks (CANs) for the grid code [18, 19, 20] are one of the few canonical models in computational neuroscience. That we find RNNs trained to perform dual agent path integration develop structure distinct from CANs is noteworthy. There are several possibilities for reconciling our results with their differences to CANs, two of which we discuss below. First, the relative space encoding that we find could be coupled with a CAN so that the resulting network path integrates the other agent by first path integrating the self and then updating the other's position estimate by using the relative spacing encoding. This kind of egocentric framework is consistent with the presence of object vector cells in MEC [58]. Second, the grid code could be one of several codes that enables path integration, and scenarios (such as the presence of another) could lead to the reduction of the grid code for another, more suitable code. This is in-line with the negative correlation between strength of grid responses and success in a multi-agent task [10].

We hope that our work inspires new experiments to uncover the representations used by the brain to enable success in multi-agent environments.

## Acknowledgments and Disclosure of Funding

We thank Cash Costello, Andy Alexander, Michael Goard, Kechen Zhang, Will Dorrell, and members of the Goard Lab for helpful discussion surrounding this work. We thank Chris Ratto, Nora Wolcott, Adele Meyers, Anna-Lena Krause, and Luís Pereira for feedback on this manuscript. WTR acknowledges funding by an Internal Research and Development grant from JHU/APL. FA acknowledges funding from the National Science Foundation, grant 2134241. NM acknowledges funding from the National Science Foundation, grant 2313150.

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

# A    Multi-agent path integration

Elite sports players are often remarkably capable of keeping track of their own movements and the movements of others. For instance, soccer/football players are able to accurately pass the ball, while running, to a teammate who is also running. While these are individuals that have trained in specialized settings for years, the "average" person has every day events that require the navigating in environments with moving agents. As an example, changing lanes on the highway requires maintaining an understanding of where one's car is, as well as other nearby cars.

While these experiences highlight that humans are able to accurately keep track of the trajectories of themselves and others, they do not definitively demonstrate that we are simultaneously performing dual agent path integration. We know of no such work that has directly addressed this question, however we hypothesize that path integrating multiple agents is possible. We consider two characteristics of path integration: 1) the ability to update an estimate of location without direct sensory inputs (e.g., maintaining an accurate estimate of position in the dark); 2) the ability to take direct routes. We argue that there is anecdotal and experimental evidence supporting that both of these can occur in dual agent settings. First, if we turn around after having passed someone in the hallway several seconds previously, we may surprised where they are (e.g., they are all the way down the hall, they are almost exactly where they were when we saw them). This suggests that we are maintaining an estimate of where another individual is through an estimation of their movement, all without direct visual inputs. And second, rats trained to perform pursuit of a laser pointer showed the ability to take "shortcuts" that were predictive in nature for specific types of laser pointer trajectories [76]. This suggests that the rats are able to leverage dynamical predictions of where the other is going.

If dual agent path integration is performed in the brain, what areas might be actively supporting the computations involved? We hypothesize that the medial entorhinal cortex (MEC) plays an important role. Both the hippocampus and anterior cingulate have been shown to be modulated by the presence of others [11, 12, 13, 14, 15, 16, 17]. Given the inter-connectivity between MEC and these brain regions, it seems likely that the MEC is also modulated. Given that the MEC is believed to perform single agent path integration, we expect that the modulated activity may reflect its role in path integrating multiple agents. In support of this, Stangl et al. (2021) [9] found that the MEC had similar border representations for both the self and other. However, because of the lack of experimental work in this direction, we cannot rule out that other brain areas are also (or primarily) involved in dual agent path integration.

Finally, we note that grid cells, and path integration more broadly, have increasingly been recognized as being computational mechanisms that are used for more than purely physical location [79]. The ability to update estimates of location in abstract spaces (e.g., frequency space [81]) suggests other evidence for being able to perform dual agent path integration. For instance, musicians can simultaneously play two melodies. Thus, future work may probe this capability in settings involving experimental variables beyond space.

Because our dual agent RNNs, which assume the existence of dual agent path integration, make predictions that can be used to test this hypothesis, we believe they will be useful in guiding experiments that probe the nature of multi-agent spatial coding.

# B    RNN architecture and training details

Code used for training and evaluating RNNs on path integration was pulled from `https://github.com/ganguli-lab/grid-pattern-formation`. Vanilla RNNs, with parameters set to that reported in Table S1 (unless otherwise noted), were trained on single or dual agent path integration. Code used for this paper is publicly available at `https://github.com/william-redman/Not-So-Griddy`.

In Sec. 2, we listed the four modifications that were made in enabling RNNs to learn dual agent path integration. Here we provide more details on each of these modifications:

**Additional inputs:** Trajectories for both agents were sampled using the same generative function [28, 29]. Each time step is characterized by movement direction, $\theta(t)$, and speed, $v(t)$. We expand the architecture of the vanilla RNN to allow for four inputs (agent 1 movement direction, agent 1 speed, agent 2 movement direction, and agent 2 speed) by setting the input variable `input_size` of `torch.nn.RNN` to $4$.

**Summed place cell activations:** The output layer of the RNN model comprises of "place cells" [37], whose "ground truth" activations are tied to the distance of an agent to specific locations ("place fields"). In-line with previous work, we use a difference-of-Gaussian model for the place field responses [29]. The ground truth activations are used as an error signal for training the RNN model in a supervised fashion. To generate the ground truth activations for the dual agent setting, we evaluate the distance of each agent to all place fields. This is used to generate activations for each place cell (for each agent, independently). We sum these responses, for each cell, to generate the "ground truth" activations (see Appendix B.1 for more rationale on this choice).

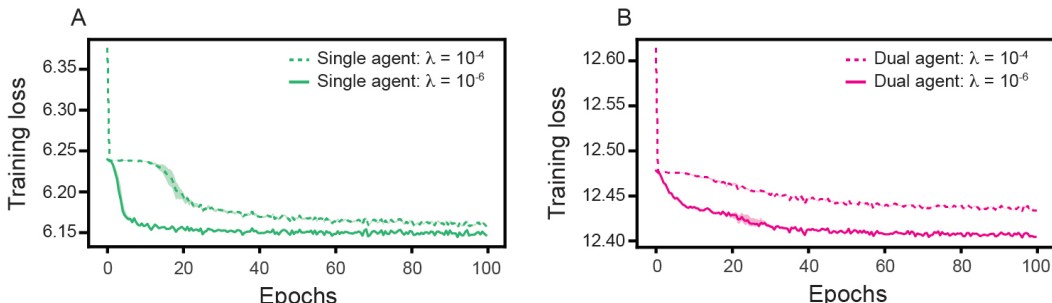

Figure S1: **Training loss for RNNs trained on single and dual agent path integration.** (A)–(B) Training loss, as a function of training epoch, for RNNs trained on single and dual agent path integration, respectively. Solid line is mean across 5 independently trained networks. Shaded area is maximum and minimum of all 5 networks.

$k$**-means clustering for decoding:** For single agent path integration, previous work has decoded the location of the agent by considering the $n_d = 3$ most active place cells and averaging the $x-$ and $y-$coordinates of their place field centers. To decode positions from place cell activations in the dual agent setting, we first consider the $2n_d = 6$ most active place cells. We then take the $x-$ and $y-$coordinates of the corresponding place field centers and apply $k-$means clustering, using the function `kmeans` from the package `scipy.cluster.vq`, setting $k = 2$. The centroids of the clusters are used as the predicted positions for the two agents. We compute the decoding error as the Euclidean distance between the true and predicted positions of the agents (the positions considered as $4-$dimensional vectors). Because the order of the centroids is not guaranteed to match the arbitrary ordering of the agents' positions, we find the ordering that minimizes the distance. As the $k-$means clusering incurs additional computational costs, we only perform the decoding twice each epoch (at the beginning and half-way through).

**Reduced weight decay regularization:** Previous work has introduced weight regularization to the weights in the recurrent layer, $\mathbf{W}^{\text{rec}}$, to encourage sparsity and aid generalization (Eq. 3 [32]). Given that we expect dual agent path integration to be a more challenging task, we hypothesize that the weight decay value used for single agent path integration might lead too much sparsity to perform the dual agent path integration task well. We therefore reduced the value used for weight decay, taken as an input in the Adam optimizer `torch.optim.Adam`, from $\lambda = 10^{-4}$ to $\lambda = 10^{-6}$.

Table S1: **Parameters used to train RNNs on single and dual agent path integration.** Unless otherwise noted (in the text), these are the values the parameters used to train single and dual agent RNNs take. Parameters that take values that are different from the defaults of previous models [29, 32] are bolded. Any values not listed are the same as they were in previous work [29, 32] .

| Parameter | Value |
| --- | --- |
| Epochs | 100 |
| Batch size | 200 |
| Batches per epoch | 1000 |
| Path length ($T$) | 20 |
| Arena length (L) | 2.2 m |
| Learning rate | $10^{-4}$ |
| Place cells ($n_P$) | 512 |
| Grid cells ($n_G$) | 4096 |
| $\sigma_1$ | 0.12 |
| $\sigma_2$ | 0.24 |
| Activation | ReLU |
| **Weight decay** | $10^{-6}$ |
| Optimizer | Adam |

The training loss associated with single and dual agent path integration is presented in Fig. S1. We note that there is no equivalent to "test" loss, as the trajectory used to train each iteration was sampled randomly from a distribution [29]. No trajectories were separately maintained in a "test set".

All training experiments and analysis were performed using an AWS GovCloud g4dn.xlarge instance with 1 GPU and 4 CPUs.

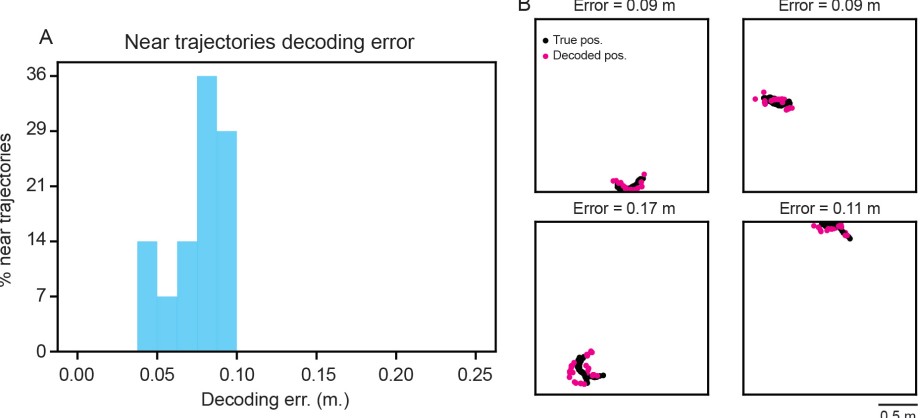

Figure S2: **Dual agent RNNs are able to decode trajectories when the two agents are near each other.** (A) Distribution of decoding error of dual agent RNN for trajectories where the two agents are on average $< 0.10$ m from each other. (B) Example true and decoded positions for the four trajectories with the smallest average distance between agents.

## B.1 Why summed place cells?

When deciding on how to define $\mathbf{p}$, the "true" place cell activity, in the dual agent setting, we considered two important experimental and model based constraints:

- **Zero mean input:** Previous work considered the DoG functional form for $\mathbf{p}$ because it maintains a zero mean across space [29]. This was motivated by earlier work that showed how non-negative PCA could learn grid cell like responses from place cell input that was assumed to be zero mean [86]. Given that prior observations of the RNN model failing to develop grid cells [33] came in part because of architectural decisions that were not supported by the original theory [59], we sought to ensure that our definition of $\mathbf{p}$ maintained "zero mean-ness" so that the lack of grid-ness we observe could not be due to our choice of $\mathbf{p}$. Note that this rules out the use of certain non-linearities, such as taking the maximum value of the place responses evaluated for each agent independently and functions that saturate (e.g., sigmoid).

- **Mixed selectivity:** Neural recordings have identified multiple variables of self and other being encoding in the hippocampus and anterior cingulate being represented by *individual* neurons. For instance, Zhang et al. (2024) [17] found that there were more CA1 neurons that were tuned to the location of self and other than there were tuned to the location of only the other. Similarly, Yoo et al. (2021) [15] found the majority of neurons encoded information about self and other's position during a virtual pursuit task. Thus, while these results do not rule out the possibility of a subpopulation of "pure" social place cells that respond only to the position of others [87], modelling spatial representations in the multi-agent setting should include mixed selectivity.

Because of these two constraints, zero mean input and mixed selectivity, we chose to model $\mathbf{p}$ as the linear sum of the DoG responses applied to the location of each agent separately. That is, $p_i(t) = \text{DoG}[\mathbf{z}_1(t), \mathbf{c}_i] + \text{DoG}[\mathbf{z}_2(t), \mathbf{c}_i]$. Note that this assumption of linearity and complete mixed selectivity is almost certainly an oversimplification, however we believe that it is a valuable place to start in this first work on exploring the kinds of representations that are useful in dual agent path integration. We believe future work exploring how inclusion of non-linearities and the existence of both disjoint and mixed place cell responses further shapes the representations that emerge in path integrating RNNs in multi-agent environments.

## C  Generalization experiment details

Let $\mathbf{W}^{\text{in}}$ denote the weights projecting from the inputs to the units in the recurrent layer. Given the difference in input dimension between single and dual agent RNNs , modifications to $\mathbf{W}^{\text{in}}$ is necessary to enable an RNN trained on single agent path integration to be tested on dual agent path integration (and vice versa). Going from single to dual agent path integration (Fig. 3A), we concatenate $\mathbf{W}^{\text{in}}$ with itself, forming a new $\mathbf{W}^{\text{in}} = [\mathbf{W}^{\text{in}}, \mathbf{W}^{\text{in}}] \in \mathbb{R}^{n_G \times 4}$. Going from dual to single agent path integration (Fig. 3B), we remove the weights corresponding to the third and fourth inputs. That is, if $\mathbf{W}_{\text{in}} = [\mathbf{W}_1^{\text{in}}, \mathbf{W}_2^{\text{in}}, \mathbf{W}_3^{\text{in}}, \mathbf{W}_4^{\text{in}}]$, where $\mathbf{W}_i^{\text{in}}$ corresponds to the weights projecting from input $i$ to all units in the recurrent layer, then we generate a new set of weights

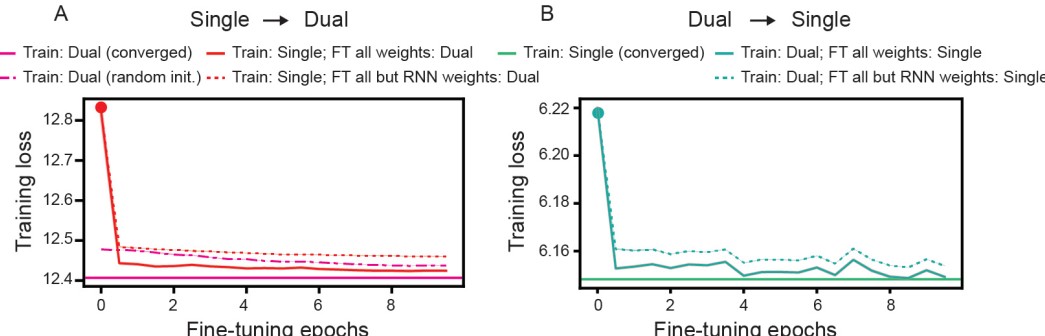

Figure S3: **Generalization trends are maintained when considering training loss instead of decoding error.** Same as Fig. 3, but when plotting training loss. Note the difference in scales is due to the fact that the dual agent RNN has summed ground truth place cell activity.

$\mathbf{W}^{\text{in}} = [\mathbf{W}_1^{\text{in}}, \mathbf{W}_2^{\text{in}}] \in \mathbb{R}^{n_G \times 2}$. Because all other training and architecture hyper-parameters are the same between RNN models trained on single and dual agent path integration, no other modifications were considered.

To freeze the weights in the recurrent layer, we set those parameters to have `requires_grad = False`.

## D  Single unit representation details

### D.1  Constructing rate maps

The construction of rate maps, which visualize activity of individual neurons , as a function of physical location, is a standard tool used to understand the neural representations underlying spatial navigation. In the context of RNNs trained on single agent path integration, this tool has been used to discover grid, border, and band cells [28, 29, 30, 31, 32]. To do this, $n_t$ trajectories, corresponding to the paths of a single agent, are generated and used as inputs to RNN models. The activations of each unit in the recurrent layer and the position of the agent at each time point is saved. The average activation value, for a given position, is then computed and plotted. Position is resolved at a resolution of $m$ bins in the $x-$ and $y-$direction, leading to $m^2$ bins in the rate map.

For the dual agent setting, we similarly sample $n_t$ trajectories, use them as inputs to dual agent RNNs, and save the activations. The primary difference for constructing rate maps for dual agent RNNs is that only the position of one of the agents is used to construct the maps (and not both). By default, we take the first agent's position. However, since there is no difference between the agents, this choice should not have any major effect.

To better understand the extent to which this choice for visualizing the rate maps affects our interpretations, we also compute rate maps where only a single agent's trajectory is sampled. That is, we trained an RNN on dual agent path integration, but then sample only a single agent's trajectory when constructing the rate maps, setting the inputs corresponding to the movement direction and speed of the second agent to $0$ (we refer to this approach as "Dual agent RNN single agent rate map"). We find that, in general, this leads to similar rate maps (compare Fig. 4B with Fig. S5B).

For all rate maps, we set $n_t = 1000$ and $m = 20$. We note that it is typical for rate maps to be smoothed by applying a filter after their construction. However, we chose not to smooth the rate maps (for either single or dual agent path integration) to ensure that the interpretations of the representations we studied were being driven by filtering.

### D.2  Functional class metrics

Visual inspection of rate maps have led to the identification of subsets of neurons and hidden units that have distinct properties (e.g., "grid-like"). To quantify these properties and enable unsupervised identification, metrics have been developed. These are discussed in detail below:

**Grid score:** Ideal grid cells are characterized by having firing fields distributed along the vertices of a triangular lattice [5, 6]. For neural data, the grid signature has been found to be more strongly apparent when considering not the rate map, but the spatial autocorrelogram (SAC) of the rate map. Let $\mathbf{X}_i$ denote the rate map corresponding to unit $i$ in the recurrent layer, and $\mathbf{S}_i$ denote its SAC. If $\mathbf{S}_i$ has strong triangular lattice structure, rotating it by $60°$ (which we denote by $\mathbf{S}_i^{60}$) should lead to a SAC that is highly correlated with the $\mathbf{S}_i$. This is similarly true for rotations of $120°$ and $240°$. High correlations could also arise when there is a finer symmetry. To identify when this is the case, the correlation between $\mathbf{S}_i^{30}$, $\mathbf{S}_i^{90}$, and $\mathbf{S}_i^{150}$ can be taken. The grid score, $\text{gs}_i$, is computed

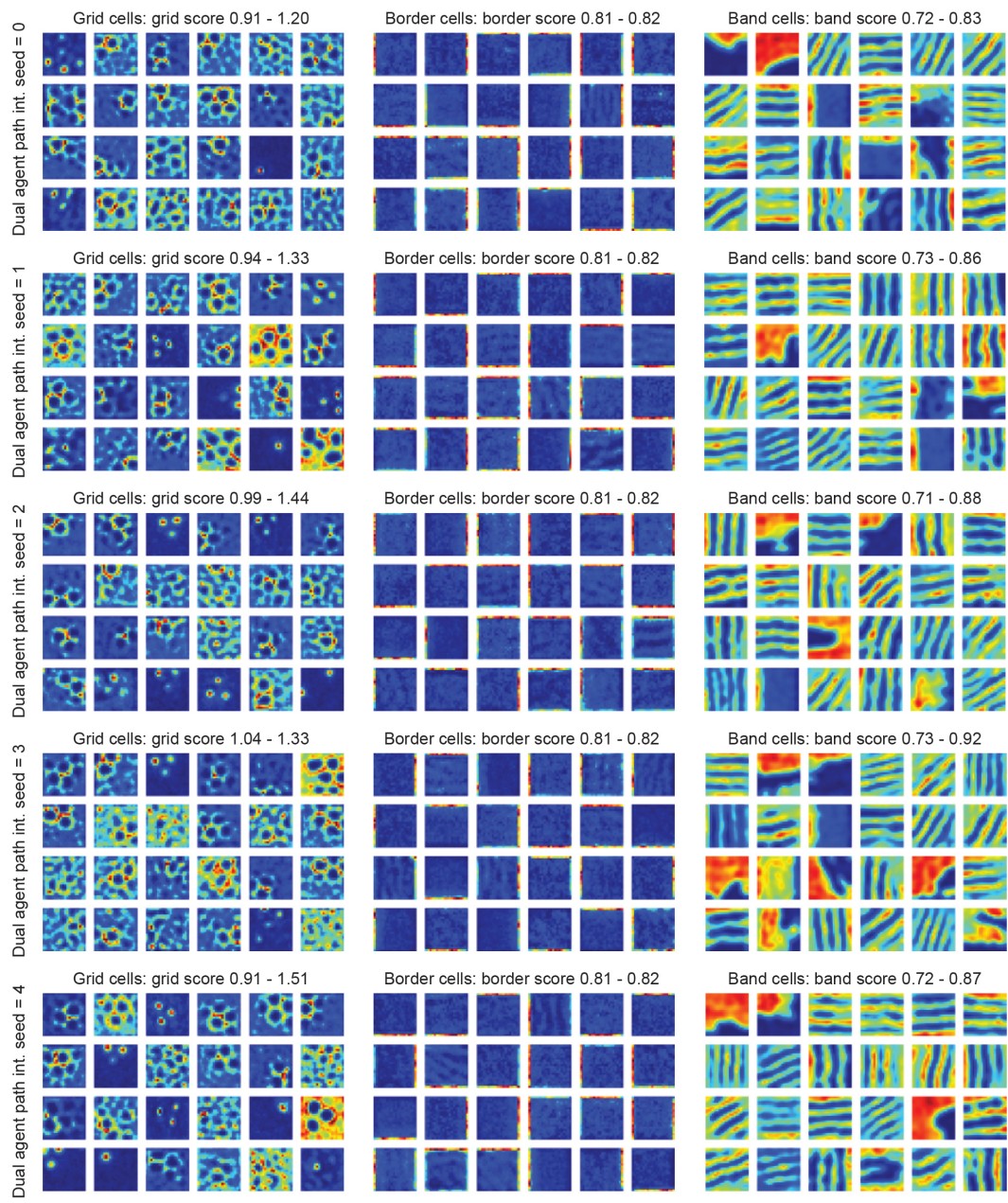

Figure S4: **Consistent individual unit level representations across independently trained dual agent RNNs**. Same as Fig. 4B (top), for all 5 seeds (each corresponding to an independently trained RNN).

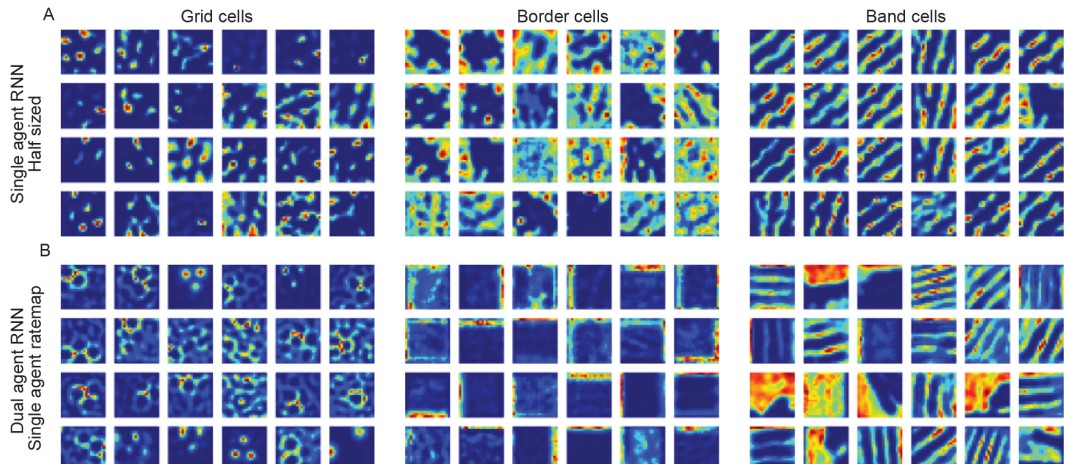

Figure S5: **Control experiment rate maps.** (A) Visualization of rate maps from an RNN with half the number of recurrent and output units trained on single agent path integration, with the highest grid, border, and band scores. (B) Same as Fig. 4B (top), but for rate maps computed from the trajectories of a single agent (Appendix D.1). Note that the RNN was, as in the case of Fig. 4B (top), trained on dual agent path integration.

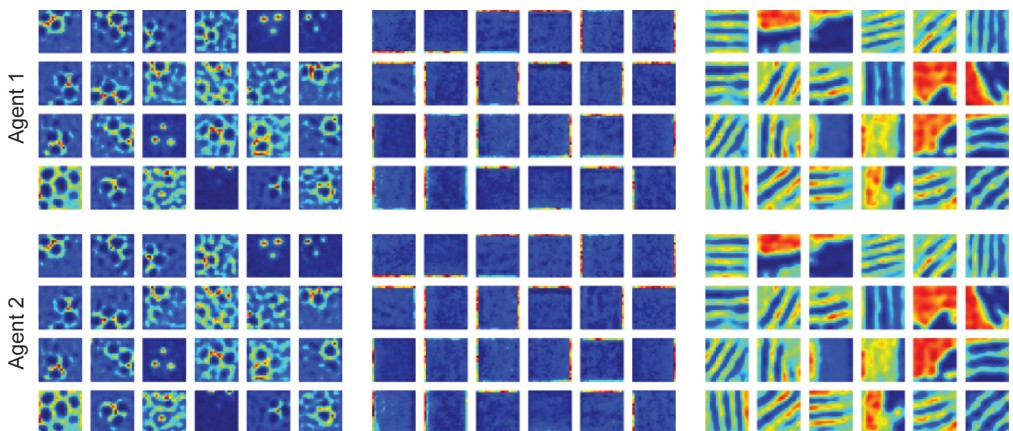

Figure S6: **Rate maps are consistent when using different agent's positions.** Top row, same as Fig. 4B in the main text. Bottom row, same as top row, but using the second agent's position to plot the rate maps, instead of the first agent's.

as

$$\text{gs}_i = \frac{1}{2}\left[\text{corr}(\mathbf{S}_i, \mathbf{S}_i^{60}) + \text{corr}(\mathbf{S}_i, \mathbf{S}_i^{120})\right] - \frac{1}{3}\left[\text{corr}(\mathbf{S}_i, \mathbf{S}_i^{30}) + \text{corr}(\mathbf{S}_i, \mathbf{S}_i^{90}) + \text{corr}(\mathbf{S}_i, \mathbf{S}_i^{150})\right], \quad (5)$$

where $\text{corr}(\mathbf{S}_i, \mathbf{S}_i^\theta) = (\sum_j \tilde{S}_{i,j} \tilde{S}_{i,j}^\theta)/\sum_j (\tilde{S}_{i,j})^2$, with $\tilde{S}_{i,j}$ being the $j^{\text{th}}$ bin of the SAC of hidden unit $i$, with the center peak masked out.

**Border score:** Ideal border cells are characterized by having their activations selectively aligned along a border. To compute the border score [7], the rate map corresponding to unit $i$ in the recurrent layer, $\mathbf{X}_i$, is first thresholded, with any spatial bin having activation $< 0.3 \cdot \max(\mathbf{X}_i)$ being set to 0. Connected components of this thresholded rate map are found using `scipy.ndimage` and those with area $< 200$ cm$^2$ are additionally set to 0 [7]. Let $\tilde{\mathbf{X}}_i = \mathbf{X}_i \odot \mathbf{M}_i$ correspond to this masked out rate map, where $\mathbf{M}_i \in \{0, 1\}^{m \times m}$ is the mask and $\odot$ is the element-wise multiplication operation. The maximum coverage along any of the four environment walls, $c_{\mathbf{M}_i}$, is computed by finding the number of non-masked out spatial bins along each wall (i.e., $\mathbf{M}_i(x, y) = 1$) and dividing it by $m$. Finally, the mean firing distance, $d_{\mathbf{M}_i}$, is computed for each remaining connected component is by taking the mean of the distance of each spatial bin in the connected component to the nearest wall, weighted by the normalized activation at that spatial bin. $d_{\mathbf{M}_i}$ is then defined by taking the average value across all connected components. The border score, bos$_i$, is the computed as

$$\text{bos}_i = \frac{c_{\mathbf{M}_i} - d_{\mathbf{M}_i}}{c_{\mathbf{M}_i} + d_{\mathbf{M}_i}}. \quad (6)$$

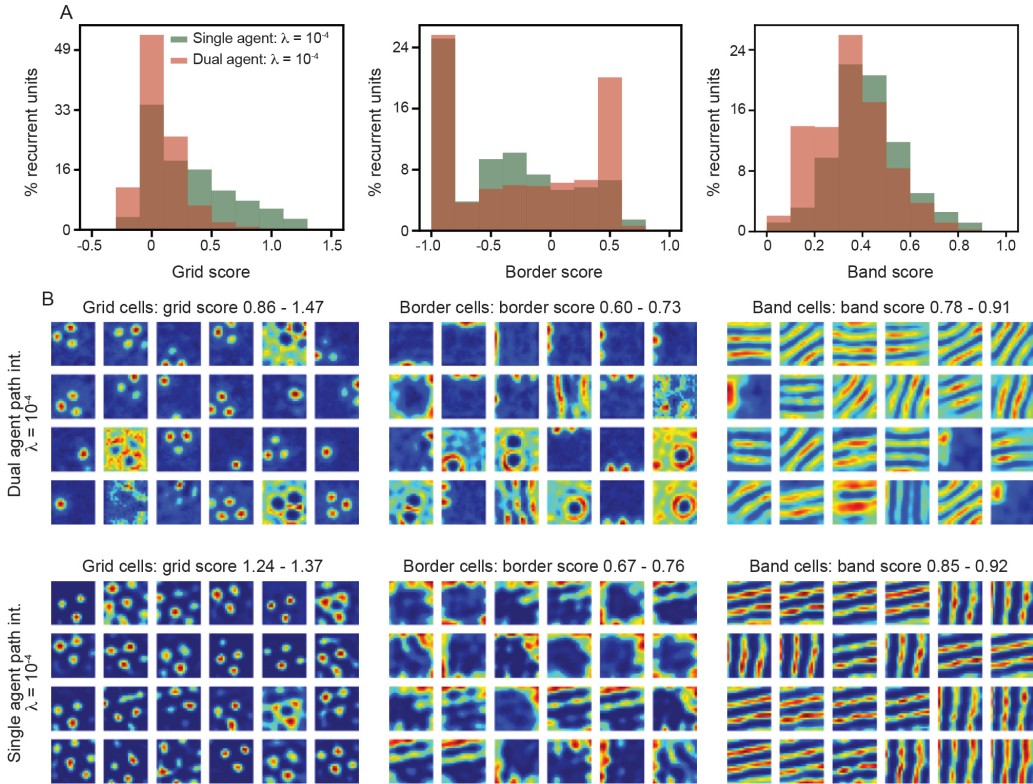

Figure S7: **Difference in functional classes that emerge in single and dual agent RNNs persists when training using greater weight decay.** Same as Fig. 4 (A)–(B), but for RNN model trained with weight decay $\lambda = 10^{-4}$. Results computed from 3 independently trained RNNs.

**Band score:** To the best of our knowledge, no metric that explicitly quantifies the "banded-ness" of neural activity or hidden unit activations exists. We therefore created one to enable the exploration of how adding an additional agent affects the development of units with band structure in their rate maps. As with grid and border scores, let $\mathbf{X}_i$ denote the rate map corresponding to unit $i$ in the recurrent layer. Let $\mathbf{X}(k_x, k_y)$ be a two-dimensional sinusoid with frequency $k_x$, in the $x-$direction, and $k_y$, in the $y-$direction. Finally, let $K = \{0, 0.1, ..., 2.0\}$ be possible values for $k_x$ and $k_y$. The band score, $\text{bas}_i$, is computed as

$$\text{bas}_i = \max_{k_x, k_y \in K} \texttt{corrcoef}\left[\mathbf{X}_i, \mathbf{X}_i(k_x, k_y)\right], \tag{7}$$

where `corrcoef` is `numpy`'s Pearson product-moment correlation coefficient.

### D.3 Functional class ablation experiments

To probe how units in the recurrent layer are used in the computations underlying the ability to path integrate one or two agents, we perform functional class targeted ablation experiments (Fig. 4C). To do this, we rank the units based on their functional properties (grid, border, or band score). We then ablate units with the highest $p\%$ of the scores by setting all incoming and outgoing weights to 0. That is, if unit $i$ is determined to have a sufficiently high score and is to be ablated, we set $\mathbf{W}^{\text{rec}}_{i,j} = \mathbf{W}^{\text{rec}}_{j,i} = 0$, for all $j = 1, ..., n_G$. Here, $\mathbf{W}^{\text{rec}}$ are the weights in the recurrent layer.

## E  Relative position representation details

### E.1  Relative position rate maps

To construct the traditional rate maps used to visualize grid, border, and band cells (Fig. 4, Appendix D.1), the $x-$ and $y-$coordinates of the environment were split into bins and the average activity, per spatial bin, was computed using one of the agent's $x$- and $y$-position.

To examine whether there was any tuning in the relative position space, we construct rate maps using not $x$ and $y$ but $\Delta_x = x_1 - x_2$ and $\Delta_y = y_1 - y_2$, where $(x_1, y_1)$ and $(x_2, y_2)$ are the locations of the two agents (the

Table S2: **Functional class ablation statistics.** Mann-Whitney test p-values for RNN model decoding error (Fig. 4C). Statistics, computed using `scipy.stats.mannwhitneyu`, correspond to testing the null hypothesis that the decoding error of the RNN model with functional classes ablated is not greater than the RNN model with random units ablated. Bolded $p$-values denote those less than 0.05.

| % units removed | Functional class | Single agent $p$-value | Dual agent $p$-value |
|---|---|---|---|
| 5 | Grid cells | 0.11 | 0.99 |
| 10 | Grid cells | $\mathbf{6.4 \cdot 10^{-4}}$ | 0.87 |
| 15 | Grid cells | $\mathbf{2.0 \cdot 10^{-3}}$ | 0.98 |
| 20 | Grid cells | $\mathbf{3.4 \cdot 10^{-6}}$ | 0.98 |
| 25 | Grid cells | $\mathbf{1.3 \cdot 10^{-2}}$ | 1.0 |
| 5 | Border cells | $\mathbf{2.9 \cdot 10^{-7}}$ | 0.24 |
| 10 | Border cells | $\mathbf{2.9 \cdot 10^{-7}}$ | 0.98 |
| 15 | Border cells | $\mathbf{2.9 \cdot 10^{-7}}$ | $8.0 \cdot 10^{-2}$ |
| 20 | Border cells | $\mathbf{2.9 \cdot 10^{-7}}$ | $\mathbf{1.4 \cdot 10^{-2}}$ |
| 25 | Border cells | $\mathbf{2.9 \cdot 10^{-7}}$ | 0.90 |
| 5 | Band cells | $\mathbf{2.9 \cdot 10^{-7}}$ | $\mathbf{4.6 \cdot 10^{-2}}$ |
| 10 | Band cells | $\mathbf{2.9 \cdot 10^{-7}}$ | $\mathbf{2.9 \cdot 10^{-2}}$ |
| 15 | Band cells | $\mathbf{2.9 \cdot 10^{-7}}$ | $9.0 \cdot 10^{-2}$ |
| 20 | Band cells | $\mathbf{2.9 \cdot 10^{-7}}$ | 0.25 |
| 25 | Band cells | $\mathbf{2.9 \cdot 10^{-7}}$ | 0.82 |

ordering is arbitrary, but fixed). This provides a transformation from the allocentric reference frame to the relative space reference frame (Fig. 5A). Note that this transformation is not one-to-one, as multiple configurations of agents position in allocentric space have the same relative space coordinates. Note that relative space reference frame has twice the range, in each dimension, as allocentric space. This is because, while $x, y \in [-L/2, L/2]$, $\Delta_x, \Delta_y \in [-L/2, L/2]$.

Interpreting the rate maps constructed in the relative space can be slightly un-intuitive. To aid in this, we provide illustrations of three relative space rate maps, with specific locations of the rate map visualized in the allocentric reference frame (Fig. S8B). As noted above, because the mapping is not one-to-one, these are not the only possible realizations of the allocentric space. From these examples, we see how a unit in the recurrent layer that is identified from its allocentric rate map as a grid cell has the same average activity across a wide range of relative positions of the two agents (Fig. S8A, left). In contrast, the example border cell responds only when both agents are along the borders (Fig. S8A, middle), which can be seen by the activations being restricted to when $\Delta_x$ is at its minimal and maximal values. For an example band cell, again there is specificity in the response related to the value of $\Delta_x$, with activity occurring only at certain intervals (Fig. S8A, right). In between those intervals, there is little activation.

## E.2 Spatial information

To quantify the amount of information neural responses encode about specific variables, the metric spatial information was developed [43]. In particular, the spatial information, $\mathrm{SI}_i$, encoded by the rate map corresponding to unit $i$ in the recurrent layer, $\mathbf{X}_i$, is computed by

$$\mathrm{SI}_i = \frac{1}{\bar{\mathbf{X}}_i} \sum_{x,y} p(x,y) \mathbf{X}_i(x,y) \log_2 \left( \frac{\mathbf{X}_i(x,y)}{\bar{\mathbf{X}}_i} \right), \tag{8}$$

where $\bar{\mathbf{X}}_i$ is the mean value of rate map $i$ over all space, $x$ and $y$ correspond to the $m$ spatial bins in each direction that are used to construct the rate map, and $p(x,y)$ is the probability that the agent was in the spatial bin $(x,y)$.

We compute the spatial information on the relative rate maps (Figs. 5, Appendix E.1). While a useful tool, the spatial information can be skewed by units that (spuriously) respond highly to only a single spatial bin (see Fig. 5E, top left rate map). Additionally, the multiplicative dependence on $p(x,y)$ can lead to high spatial information in specific bins, when the probability is skewed. Indeed, we find that $p(x,y)$, in the relative space, is heavily distributed towards the center of the rate maps, corresponding to the two agents being near each other. Thus, while a useful tool for enabling us to identify a new functional class of units, *these results do not rule out other interesting responses in relative space (or other reference frames)*.

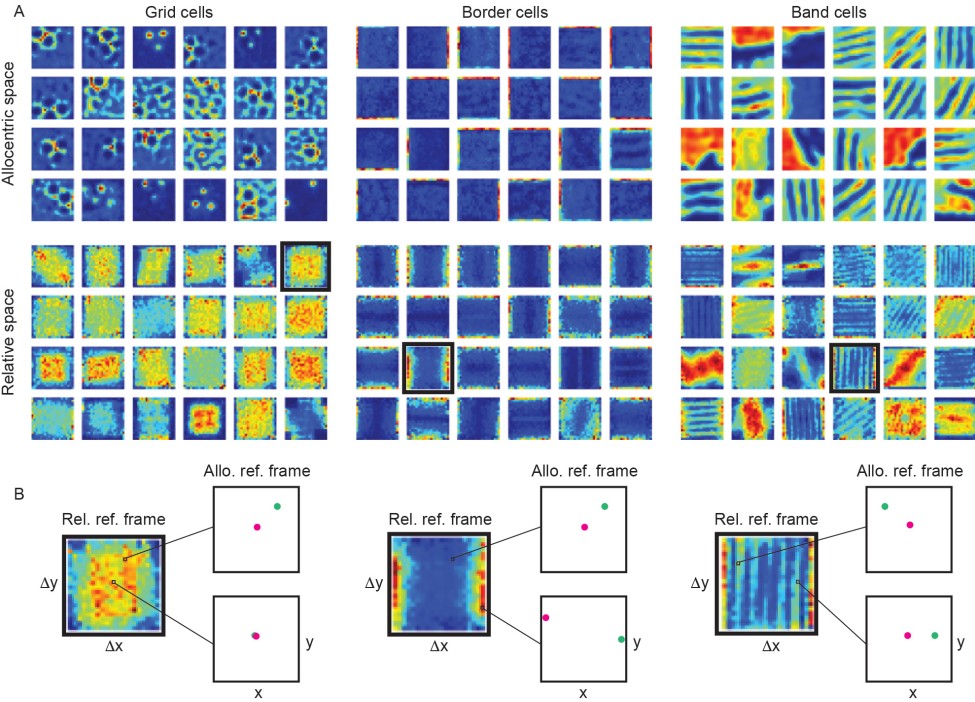

Figure S8: **Border and band cells additionally encode relative space information, but not grid cells.** (A) Comparison of rate maps in allocentric space (top) and relative space (bottom), for units, from an individual dual agent RNN, that had high grid, border, and band scores. Top row is the same as Fig. 4B (bottom). (B) Example relative space rate maps (with the functional class their allocentric ratemaps had high scores for), with schematic illustrations of possible configurations for different locations in relative space.

Table S3: **Relative space ablation statistics.** Mann-Whitney test p-values for dual agent RNN model decoding error (Fig. 5E) for units with the highest relative space spatial information ablated (Appendix E.3). Details same as Table S2.

| % units removed | $p$-value |
|---|---|
| 5 | $\mathbf{1.3 \cdot 10^{-3}}$ |
| 10 | $\mathbf{6.4 \cdot 10^{-4}}$ |
| 15 | $\mathbf{6.8 \cdot 10^{-3}}$ |
| 20 | $\mathbf{4.6 \cdot 10^{-4}}$ |
| 25 | $\mathbf{2.2 \cdot 10^{-3}}$ |

## E.3 Relative space ablation experiments

To probe the importance of the units in the hidden recurrent layer that encode relative space, we perform the same ablation studies as described in Appendix D.3, but ranking units based on the spatial information present in their relative space rate maps, as opposed to their grid, border, or band scores.

## F Population representation details

### F.1 Topological data analysis

TDA [44] has become a widely used tool in neuroscience [25, 45, 46, 47, 48, 49]. In particular, it has been leveraged to provide evidence for the population activity of grid cells (from the same module) being constrained to a two-dimensional toroidal manifold [25]. A core tool in TDA is persistence (co)homology, which is used to identify topological structure present in data. Persistence (co)homology, applied to neural activity, can loosely be understood by the following overview. First, the activity of all $N$ neurons, at each time point, is viewed as a point in the space of all possible population activations. This leads to a point cloud in $N$-dimensional space. Balls, of dimension $N$, are then centered on each point in the cloud. Gradually, the radii of these balls is

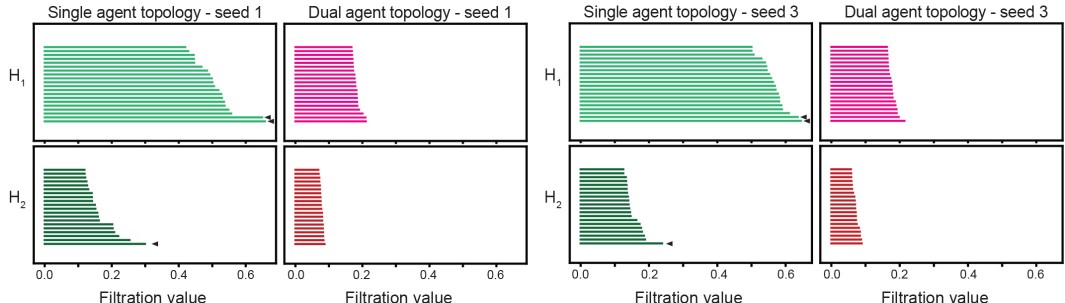

Figure S9: **Difference in topology of population level activations corresponding to single and dual agent RNNs is robustly observed across independently trained networks.** Same as Fig. 6A), for two additional example persistence diagrams, corresponding to two other independently trained RNNs.

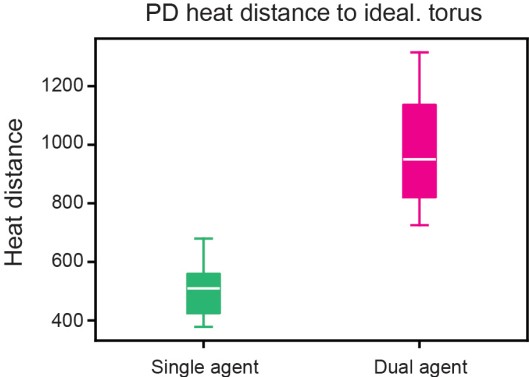

Figure S10: **Single agent persistence diagrams are closer to idealized torus than dual agent persistence diagrams.** Box plot corresponding to heat distance between the persistence diagrams of the single and dual agent RNNs, with the persistence diagram of an idealized torus.

increased. When two balls intersect, they are considered a connected component. The topological properties of all connected components is then computed. In particular, the existence of loops ($H_1$) and two-dimensional cavities ($H_2$) are extracted and the persistence of these features is measured. This can be visualized in a format called a persistence diagram.

We computed the Vietoris-Rips persistence diagram corresponding to the activations of all recurrent units for both single and dual agent RNNs. To do this computation, we used the `Ripser` python package [88, 89]. However, persistence diagrams can be subject to noise and are more challenging to compute for high-dimensional spaces. Therefore, to ensure that the differences in topology observed between single and dual agent RNNs was due to different properties underlying the population activations, and not noise, we projected the population activations onto a lower-dimensional subspace and re-computed the persistence diagrams. There are several possible choices for reducing the dimensionality. A natural option is principal component analysis (PCA). However, we found that, for the single agent RNN population activations, the subspace identified by PCA does not enable good multi-output regression with the agent's location Using partial least squares (PLS) [90], which has been successfully applied in neuroscience applications [91], we found that both single and dual agent RNN population activation, captured in a 10-dimensional subspace, can achieved high multi-output regression with the agents' positions. Therefore, we use PLS to reduce the dimensionality of the population activations, and then computed the persistence diagram on this reduced subspace (Fig. 6). The PLS was computed using `PLSRegression` from `sklearn.cross_decomposition`. Additional examples of persistence diagrams, corresponding to the activations of single and dual agent RNNs, are shown in Fig. S9.

To understand how the topology of the population level activations of the single and dual agent RNNs evolves with training, we computed the persistence diagram associated with an "ideal" two-dimensional torus and measured the topological distance [51] between the observed persistence diagram and the ideal persistence diagram. We find that the topology of the single agent RNN is closer to an idealized torus than the topology of the dual agent RNN (Fig. S10), further supporting our conclusion that the dual agent RNN does not develop the same attractor structure as the single agent RNN.

## F.2 Dynamic similarity analysis

The evolution, in time, of the activations of units in RNNs can be viewed as a dynamical system. The features of this dynamical system (e.g., fixed points, limit cycles, quasi-periodic attractors, chaos) are dependent on the inputs and structure of the RNN. Thus, characterizing corresponding dynamical properties offers the ability to gain insight into the organization of the RNN. Unsurprisingly, use of geometric and topological methods to interrogate RNN population activation can fail to properly identify similar and dis-similar dynamics, as they are sensitive to the underlying manifold the activation evolves along [52]. Recent work [53] has used data-driven tools [92, 93, 94] from Koopman operator theory [95, 96] to extract properties of RNN dynamics that can be compared using a precise notion of equivalence, called "topological conjugacy" [97]. Similar approaches have been used to study the dynamics associated with training deep neural networks [98], as well as iterative optimization algorithms, more generally [99]. Therefore, we complimented the geometric and topological analysis performed on the structure of the population level representations (Fig. F.1) with dynamical similarity analysis (DSA) [53].

To perform DSA, $K \in \mathbb{N}$ trajectories in physical space, corresponding to $K$ trajectories in population activity space, are sampled. Each trajectory has temporal length $T \in \mathbb{N}$. All the activations are collected into a tensor, $\mathbf{X} \in \mathbb{R}^{K \times T \times n_G}$, where $n_G$ is the number of units in the recurrent layer. We then perform principal component analysis (PCA) on $\mathbf{X}$, reducing the number of dimensions so that we get a new tensor $\mathbf{X} \in \mathbb{R}^{K \times T \times n_M}$, where $n_M < n_G$. Next, $D \in \mathbb{N}$ time-delays are then applied, where $D < T$, and the resulting tensor $\mathbf{H} \in \mathbb{R}^{K \times (T-D) \times n_M D}$ is then flattened along its first dimension. Dynamic mode decomposition [100] is then performed to get a compact representation of the dynamics in terms of the Koopman operator [101], which is restricted to have rank $n_R < n_M D$. The Koopman-based representation between the dynamics of the single and dual agent RNN are then compared, via the "Procrustes over vector field" metric [53].

As can be seen in the brief overview of DSA, there are several hyper-parameters whose value must be chosen. These include $K$, $D$, $n_M$, and $n_R$. To be consistent with the geometric and topological analysis, we set $K = 1000$. Because the length of the paths ($T = 20$) is small, we cannot use too many time-delays. Therefore, we set $D = 4$. Because a small number of dimensions were necessary to capture most of the variance of the activations in both single and dual agent RNNs, we considered $n_M = 10$ and $n_R = 30$ (Fig. 6). To determine whether this choice affected the conclusion that single and dual agent RNNs have different dynamical properties, we performed the analysis again, with $n_M = 50$ and $n_R = 100$. Fig. S11 shows a similar conclusion to Fig. 6, with the Procrustes analysis over vector field metric being nearly $2\times$ larger for comparison between single and dual agent RNNs than between single and single agent RNNs or dual and dual agent RNNs. However, because more modes are kept ($n_M = 50$), more noise is present, and the difference is weaker than when fewer modes are kept.

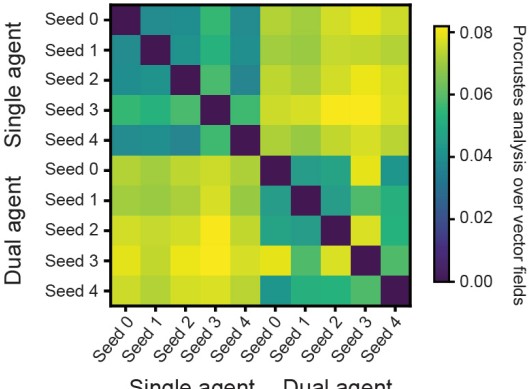

Figure S11: **Difference of activation dynamics between single and dual agent RNNs is consistent with different choice of DSA hyper-parameters.** Same as Fig. 4, but with a different choice of DSA hyper-parameters.

