# OpenReview forum: "Not so griddy: Internal representations of RNNs path integrating more than one agent"
_NeurIPS.cc/2024/Conference — NeurIPS 2024 poster_

### Official Review · Reviewer_bmf9 · 2024-07-07

**Soundness:** 3
**Presentation:** 3
**Contribution:** 2
**Rating:** 5
**Confidence:** 3

**Summary:**

This paper studies RNNs trained to path-integrate the position of two agents. They show the network behaves differently to similar networks trained to path-integrate the position of a single agent, and make some neural level predictions.

**Strengths:**

The paper was very nicely written and presented.

The question was clear, the steps taken sensible, and the results carefully discussed.

**Weaknesses:**

One big choice that seems likely to have heavily influenced the learnt representations is the choice of mixed place cell output. Surely when another animal appears in my scene I am still able to report a factorised encoding of my own position? If you make this change, and ask the network to output two factorised codes, do your results change? If so this risks becoming quite a specific point, though it would then be interesting to argue about the causes of the mixing or not mixing.

If I understood correctly, the position decoding using that k-means approach is never used to train the network, it is merely the visualise the position estimate. If this is correct, it seems a strange metric against which to compare networks, e.g. figure 3, without further comment. Rather, the key metric is the loss that is used to train the network, do the same trends hold there?

Largely, this paper seemed to say 'here's a different setting, in which things are different'. The path-integrating literature is great because we know so much about how this circuit works, and it has such a correspondence with biology. Understanding in more detail how this path-integrator you present works, for example examining the connectivity structure, or the velocity update scheme as people are able to do for traditional CAN models, would significantly improve the impact of this work, as it seems comparisons to existing neural data are not currently possible.

Personally, I agree with the critiques of the Sorscher framework, there seem to be others that match more grid cell characteristics, but its certainly a good starting point.

**Questions:**

In the 2-agent case you showed the tuning curves to one agent's position. I really wanted to know whether the tuning to the two different agents was similar in the same cell? I don't think this was reported anywhere.

**Limitations:**

The authors discuss their limitations.

---

> ### Author Rebuttal · Authors · 2024-08-05
>
> We thank the reviewer for their time and helpful comments. We are encouraged that they found our work well written and our results carefully discussed. Below we respond to the specific questions the reviewer had.
>
> **"One big choice that seems likely to have heavily influenced the learnt representations is the choice of mixed place cell output."**
>
> This is an important point, and one that has been brought up by all reviewers.  We appreciate the questions, as they helped us realize that the rationale for this choice was not clear. We have provided an answer to this in the global response (G1). If this remains unclear, we would be happy we would be happy to discuss it in more detail.
>
> Here, we specifically answer your question of factorisation. We agree that, when another agent enters the environment, we can represent our location and their location in a separable way. Given the width of the place fields we consider, this separability occurs in many of the trajectories we considered. However, in our dual agent RNN, the place cells themselves are not separable (i.e., there is no place cell for agent 1 and not for agent 2). Therefore, the dual agent RNN has to learn to properly integrate the velocity signals with the correct agent. We believe that this is a reasonable assumption, although certainly a simplification as we have more information about ourselves than others. As noted in the Discussion of our submission, we expect that if we provide more noise to the trajectory of the first agent (“the other”) than the second agent (“the self”), we will find that the dual agent RNN representation approaches that of the single agent (as only one agent’s trajectory is reliable). Exploring this trade-off would be an especially interesting future direction.
>
> **"Do the same trends hold for generalization when considering loss and not decoding error?"**
>
> We chose to focus on the decoding error as it is an easier metric, than the loss, to interpret. However, we agree that, just because there is a low decoding error, it is not guaranteed that there is a low loss. We appreciate your suggestion and have made the same plot as Fig. 3, but with the training loss (Fig. 2 in the global response pdf). We see similar trends, suggesting that our conclusions are not due to the focus on the decoding error. We will add this figure to the Appendix and discuss it in Sec. 3.1.
>
> **"Largely, this paper seemed to say 'here's a different setting, in which things are different' "**
>
> We appreciate this comment, as it emphasizes that we could strengthen the general framing of our work. To re-summarize: motivated by research finding that MEC and hippocampus are modulated by the presence of others, we asked what kinds of representations MEC could have to support multi-agent path integration.
>
> Applying a trained single agent RNN to the dual agent setting, we find that the recurrent layer must be changed in order to support dual agent path integration. This result is itself a contribution, as we are unaware of this being previously shown (although, it is perhaps not so surprising). The failure of trained single agent RNNs to be extended to the dual agent setting motivated us to develop a dual agent RNN. We find that the dual agent RNN is able to not only perform dual agent path integration, but also single agent path integration. Consequently, this argues for representations other than what are found in single agent RNNs as being optimal for multi-agent settings.
>
> To begin to shed light on what the dual agent RNN is learning, we asked: 1) what are the representations and network structure that emerge in the dual agent RNN? , and 2) how are they different from the single agent RNN? We find that the unit level representations and population level representations are distinct. However, our interest in the dual agent setting comes from it being the *setting where single agent RNNs fail*.
>
> The fact that we find no evidence for 2-d toroidal topology suggests that there may not be an underlying continuous attractor network. This makes it challenging to pursue some of the analyses that the reviewer rightfully points out as being natural. However, we tackled examining how different the dual agent RNN is from the single agent RNN by looking for grid cells in relative space. Not finding any, we concluded that the dual agent RNN learns distinct computations from the single agent RNN.
>
> We hope this re-framing of our work better demonstrates that it is not just reporting a different setting where things are different. We will integrate this motivation into our revised manuscript (Introduction and Discussion).
>
> **"I really wanted to know whether the tuning to the two different agents was similar in the same cell?"**
>
> We appreciate this suggestion. Following your comment, we have computed the tuning for all three functional classes when considering the position of the first agent or the second agent, although the ordering is arbitrary (Fig. 3 in the global responses pdf). We find the tuning curves to be almost identical (we did check that they are not exactly the same). This emphasizes that the dual agent RNN units likely do not develop tuning for individual agents. We believe this is due to our choice of summing the place cell activations, leading to no distinguishing between the agents, with respect to the loss. This is, of course, a simplification. As noted in the Discussion section, an important step in future work will be to consider what happens when one agent is made to be more representative of the self (which can be done by: including less noise in the inputs of the “self” vs. the “other”; increasing the importance of the “self’s” loss, as compared to the other, etc.). We will include this figure in the Appendix and discuss its implications in Sec. 3.2.

---

> > ### Comment · Reviewer_bmf9 · 2024-08-11
> > **Response**
> >
> > Thank you for your thorough response and engagement. Thank you for the additional plots in the rebuttal pdf, they are appreciated.
> >
> > First, on the evidence of MEC doing two agent integration: the existence of cells tuned to other agents does not imply dual-agent integration. The same single-agent integration system could just switch the agent that it is representing over time. Is there neural evidence of two agents being simultaneously represented?
> >
> > Second, there's something about the mixed-cell response that I am still not getting. Sure, if the two agents' trajectories are well-separated and correctly tracked then the two place cell outputs will be separated. But, crucially, how does the agent know which place cell bump is its own, and which is the other agent's? This seems vital, but I think the loss does not force it to be true, your decoder is ambiguous to which agent is where, and the cellular responses don't seem to distinguish the agents. If I've not misunderstood, then this is very worrying! We are certainly capable of distinguishing whether it is me or someone else who is in each position, leaving this choice of setting to be of questionable relevance for the brain?
> >
> > Third, the authors argue that many CANs path-integrating separately is not scalable. Perhaps. But it's not clear to me there are good alternatives? Surely, in some fashion, the same computations must be going on in your network, which you find struggles to train beyond 2 agents perhaps due to lack of scalability. If not, what fundamentally different thing do you think your network is doing allows it to scale (if indeed it can).
> >
> > I agree it is interesting how a neural circuit solves this problem, and I apologise for some of my review's harshness. Perhaps I have the wrong threshold for sufficiently interesting, but I think to increase my score I would need some more compelling match to neural data, or understanding of the computation this different network is doing. That is unreasonable to ask for in a rebuttal, leaving us at an impasse I'm afraid.
> >
> > Please do correct me if I have been stupid.

---

> > > ### Author Response · Authors · 2024-08-12
> > >
> > > We thank the reviewer for their continued engagement with our submission and we respect that they maintain a high bar for novel/interesting work. We believe a few additional comments may help to clarify the questions addressed by the reviewer.
> > >
> > > First, there is no neural evidence of two agents being simultaneously represented. However, we believe that there are anecdotal examples that suggest it possible (e.g., the soccer player that has to make actions based on their and others positions', the driver that has to keep track of their speed and others speeds'). Our rationale for discussing the tuning of MEC for both self and other [Stangl et al., (2021); Wagner et al., (2023)] was to suggest that *if* the brain does simultaneously represent multiple agents, a reasonable candidate for where this would occur is MEC.
> > >
> > > Second, at the start of each trajectory, the true starting position of both agents is encoded into the initial state of the RNN through the learnable weights $\textbf{W}^\text{back}$. From this initial state, the dual agent RNN must correctly integrate agent 1's velocity to update agent 1's position, and integrate agent 2's velocity to update agent 2's position. Thus, while we do not distinguish between self and other in this work, we do enforce that the RNN has some notion of which "bump" is which, as the correct inputs must be integrated to update its position.
> > >
> > > Third, in our work we have identified that some units in the dual agent RNN develop tuning for the relative position of the two agents. This enables flexible and efficient computations, such as encoding when both agents are near the borders (Fig. S5), when both agents are moving parallel to each other (Fig. S5), and when both agents are near each other (Fig. 5). We believe that the representation of relative space is useful for tasks beyond dual path integration, such as one agent pursuing another. These are not computations that would emerge in a model that featured two separate CANs.
> > >
> > > Finally, we agree that our work would be strengthened by having neural results to compare to. We hope that our results inspire experimentalists to examine MEC in multi-agent settings so that such comparisons can be done in the future. However, we note that work by Wagner et al. (2023) found weakened grid cell responses correlated with better performance in a task where one agent had to track another agent's position. Why this would be is not readily apparent, and we know of no attempts to model this. That our dual RNN model develops weakened grid responses demonstrates an alignment with existing neural data (albeit, neural data that was recorded with fMRI) and suggests that grid cells may not be optimal for representing two agents simultaneously.

---

> > > > ### Comment · Reviewer_bmf9 · 2024-08-13
> > > > **Response Response**
> > > >
> > > > Ooooo, your second point is a good point - hadn't thought of that...
> > > >
> > > > Nonetheless, it remains an issue to decide the best readout structure to use, and it worries me the extent it might be determining the claims about different path-integrating mechanisms. Evidence of neurons encoding both agents, e.g. in hippocampus, could either be outputs of a network like yours, or a system that switches which agent it is encoding, or just part of readout that is only partially overlapping in the manner you suggest but elsewhere there might be self vs. other neurons. Perhaps the bat results constrain this more. But, since all reviewers are recommending acceptance it seems like you will get in, if you wanted to increase impact on neuroscientists before publication this would be where I would sure things up - see if you conclusions generalise to completely disjoint output codes, or mixtures.
> > > >
> > > > And I take your third point, that is constraining facts about the computation going on in this RNN in interesting ways.
> > > >
> > > > And the Wagner point is interesting.
> > > >
> > > > Thanks for your engagement

---

> > > > > ### Author Response · Authors · 2024-08-13
> > > > >
> > > > > That's a fair point. We will work on exploring the impact the readout structure has and discuss this in more detail in the revised manuscript (Sec. 4).
> > > > >
> > > > > Thank you for your time and your efforts to improve our work!

---

### Official Review · Reviewer_bz9a · 2024-07-10

**Soundness:** 3
**Presentation:** 3
**Contribution:** 3
**Rating:** 6
**Confidence:** 3

**Summary:**

The authors study neural representations of space in artificial agents that perform path integration of the positions of two agents simultaneously. This is motivated by recent neuroscience studies showing place cells that respond to the location of nearby animals. The authors feed the velocity and head direction inputs of both agents into an RNN and train the network to generate the place cells. Similar to previous work with single agent position, they evaluate the spatial tuning of neurons in the RNN, specifically looking for border cells and grid cells. Unlike in the work with a single agent position, they found fewer grid cells. This indicates that representing the positions of two agents relied on a somewhat different neural code than representing the position of a single agent.

**Strengths:**

To my knowledge this is a first study that explicityly tries to model activity of the neurons in the hippocampal formation using path integration for two agents.

The paper is well written, and the authors conducted a set of ablation studies.

**Weaknesses:**

There is no attempt to analytically derive or explain the obtained result. It remains unclear whether changing some of the hyperparameters could have led to a different result. If the experiments were complemented with derivations, like in some previous work on the emergence of grid cells, that would strengthen the paper.

**Questions:**

Do authors consider a possibility biological systems might not really perform dual "path integration" (they could infer the position of other agents using visual inputs rather than using path integration)?

Is it possible that populations of the cells encoding each agent's position would be entirely or somewhat separate? Would that lead to emergence of the grid code?

**Limitations:**

Authors have adequately addressed the limitations.

---

> ### Author Rebuttal · Authors · 2024-08-05
>
> We thank the reviewer for their time and helpful comments. We are encouraged that they found our work novel and well written. Below we respond to the specific questions the reviewer had.
>
> **"There is no attempt to analytically derive or explain the obtained result"**
>
> We agree that the lack of analytical theory is a weakness of our paper. However, we believe that our numerical results, which (as the reviewer noted) are the first to study how multi-agent environments may require different kinds of spatial representations than single agent environments, open up a number of interesting questions for theorists to tackle. Therefore, we hope it wiil have a similar impact as Banino et al., (2018) and Cueva and Wei (2019), which led to a general theoretical framework by Sorscher et al. (2019).
>
> **"Might biological systems not really perform dual 'path integration' "?**
>
> This is a great question. We realized that we did not sufficiently address this point in the original submission. As all reviewers had similar questions, we have provided our response in the global comment (G2). If this remains unclear, we would be happy to discuss it in more detail.
>
> **"Is it possible that populations of the cells encoding each agent's position would be entirely or somewhat separate?"**
>
> This is an important point. We appreciate these questions as they emphasize that the rationale for this choice was not clear. We have provided an answer to this in the global response (G1). If this remains unclear, we would be happy we would be happy to discuss it in more detail.
>
> Here, we answer the specific question of what would happen if the populations of place cells encoding each agent were entirely separate. In this case, we expect that two separate 2-dimensional toroidal attractors would develop, each with grid cells. While we think that this is an interesting possibility (and one that we do not fully discount), we believe that mixed selectivity (i.e., neurons responding to multiple different features - in this case, different agents) is a crucial facet of neural computation. Consequently, we consider fully disjoint place cell representations to be unlikely in the hippocampus.
>
> **References:**
>
> Banino et al (2018) Nature “Vector-based navigation using grid-like representations in artificial agents”
>
> Cueva and Wei (2019) ICLR “Emergence of grid-like representations by training recurrent neural networks to perform spatial localization”
>
> Sorscher et al. (2019) NeurIPS "A unified theory for the origin of grid cells through the lens of pattern formation"

---

> > ### Comment · Reviewer_bz9a · 2024-08-12
> >
> > Thank you, I appreciate the clarifications.

---

### Official Review · Reviewer_rAUq · 2024-07-12

**Soundness:** 3
**Presentation:** 4
**Contribution:** 3
**Rating:** 7
**Confidence:** 4

**Summary:**

This paper studies the internal representations of recurrent neural networks that have been trained to path integrate two agents simultaneously, based on the hypothesis (and related experimental evidence) that individual agents account for the positions of others in multi-agent environments. The authors augment existing single agent vanilla discrete-time RNN models of path integration to perform the same for two agents. Through several numerical experiments on these models, the authors show that grid cell responses are weaker in the dual agent case compared to the single agent case, while border and band cell responses are stronger. Then they show that these dual agent RNNs encode information on the relative position of the two agents. Furthermore, they show that networks capable of performing dual agent path integration can generalize to the single agent task, while the reverse does not hold in practice. Finally, they outline testable predictions of their model (weaker grid responses, stronger border and band cell responses, tuning for relative position) and future directions for research on the computations underlying spatial navigation.

**Strengths:**

1. The writing is clear and convincing, and the authors have situated this work well in the context of previous research. The figures and overall presentation are great (**minor nit:** it would be better to use vector images instead of high-res PNGs, if possible). This work provides a novel perspective and is a useful contribution to the rich literature on the emergence of grid cell representations for spatial navigation.
2. The experimental analysis is straightforward and solid, with several important control experiments and metric analyses performed to test the strength of the results.
3. The authors outline several clear testable predictions to validate their results using experiments.

**Weaknesses:**

1. The multi-agent experiments are limited to dual agent path integration. Studying briefly the strengths of grid and border cell responses in the 3-, 4- and perhaps 5-agent cases would strengthen the paper, given that the code framework seems flexible. This is indeed a limitation/direction for future work that the authors themselves have acknowledged.
2. The RNNs used in the experiments are not noisy and are simple vanilla discrete-time RNNs. It would be good to study the interplay between recurrent noise and the grid and border scores. It might also be interesting to analyze continuous-time RNNs (where noise improves stability and robustness, see Lim et al., 2021 [1]).
3. Certain architectural assumptions could be justified better:
    1. The results and predictions of this work are strongly linked to those of Sorscher et al., 2021 [2], which may not be the only method by which grid cell representations emerge in a neural circuit. The underspecification of the problem means that alternative hypotheses [3,4] exist that can explain the emergence of grid cells, and these could be studied in future work.
    2. Is there any biological evidence that the same neural circuits perform path integration simultaneously for multiple agents? Could a single agent network not be used to estimate the positions of multiple agents separately and sequentially?
    3. Are there cases where the two agents are present in adjacent/extremely close locations but the $k$-means clustering does not decode the position of one agent correctly (since it always identifies 2 centers from the top-2$n_d$ active output units)? More generally, could the authors elaborate on the accuracy of path integration when the agents are close together?
    4. Have the authors considered alternatives to the "social" summed place cell activations? Perhaps having twice the number of output units instead (although, this is admittedly not as scalable to multi-agent cases)?

**References:**
1. Lim et al. "Noisy recurrent neural networks." Advances in neural information processing systems 34 (2021).
2. Sorscher et al. "A unified theory for the origin of grid cells through the lens of pattern formation." Advances in neural information processing systems 32 (2019).
3. Schaeffer et al. "Self-supervised learning of representations for space generates multi-modular grid cells." Advances in neural information processing systems 36 (2023).
4. Xu et al. "Conformal Normalization in Recurrent Neural Network of Grid Cells." arXiv preprint arXiv:2310.19192 (2023).

**Questions:**

See the Weaknesses section.

**Limitations:**

The authors have adequately discussed the limitations of the work, but could potentially elaborate on the effect of and limitations related to certain architectural assumptions (see W3).

---

> ### Author Rebuttal · Authors · 2024-08-06
>
> We thank the reviewer for their time and insightful comments. We are encouraged that they found our work well written and our analysis solid. Below we respond to the specific questions the reviewer had.
>
> **"The RNNs used in the experiments are not noisy and are simple vanilla discrete-time RNNs."**
>
> This is an interesting point. Some of the original work demonstrating that RNNs could learn grid cells through training on path integration [Cueva and Wei (2019)] found it necessary to have recurrent noise in order to see good functional classes. However, we are unaware of any subsequent RNN models (including the Sorscher et al. model) that make use of noise. Since we have focused our work on comparing directly to the Sorscher et al. model, we believe the study of noise is outside the scope of this work.
>
> However, we agree that it would be interesting to understand how noise affects border and grid scores. In particular, future work could study whether recurrent noise can lead to the variability in grid orientation and spacing that was recently reported in individual grid modules of rat MEC [Redman et al., (2024)]. We will mention these exciting avenues for future research in the revised version of the manuscript (in the Discussion) and we thank you for this suggestion.
>
> **"Is there any biological evidence that the same neural circuits perform path integration simultaneously for multiple agents?"**
>
> This is a great question and one we realized we did not sufficiently address in the original submission. As all reviewers had similar questions, we have provided a response to this in the general comment (G2). If this remains unclear, we would be happy to discuss it in more detail.
>
> **"Are there cases where the two agents are present in adjacent/extremely close locations but the k-means clustering does not decode the position of one agent correctly?"**
>
> This is a good question and we appreciate you bringing up this possibility. To answer your question, we computed the decoding error for trajectories that had a median distance between the two agents of less than 0.10 m (less than the width of the place fields). We find that the dual agent RNN performs well in this setting (Fig. 1 in the global response pdf). Thus, we do not find any cases where the k-means clustering fails. We believe this is due to the fact that we run the k-means clustering on the place field centers of the top 6 most active hidden units. Thus, as long as the top 6 place fields are near the position where both agents are, the clustering should identify locations that are near the true positions. We will add this figure to the Appendix and discuss it in Sec. 3.2. Additionally, we will note that other choices, besides the k-means clustering, could be used and may achieve better decoding.
>
> **"Have the authors considered alternatives to the "social" summed place cell activations?"**
>
> This is an important point. We appreciate this question, as it emphasizes that the rationale for this choice was not clear. We have provided an answer to this in the global response (G1). If this remains unclear, we would be happy to discuss it in more detail.
>
> **"Studying briefly the strengths of grid and border cell responses in the 3-, 4- and perhaps 5-agent cases would strengthen the paper"**
>
> We agree that studying grid and border cells in environments with more agents is an important and interesting avenue to pursue. Our initial experiments on tri-agent RNNs have not yielded strong results on the path integration on three agents. We believe this is because we have run up against the capacity of the RNN, and therefore need to modify the architecture to achieve good results. We have begun to explore the parameter space for tri-agent RNNs, but are still in the early phases. We will further emphasize that this is a limitation of our current work and highlight it as in important future direction.
>
> **References:**
>
> Cueva and Wei (2019) ICLR “Emergence of grid-like representations by training recurrent neural networks to perform spatial localization”
>
> Redman et al., (2024) bioRxiv “Robust variability of grid cell properties within individual grid modules enhances encoding of local space”

---

> > ### Comment · Reviewer_rAUq · 2024-08-10
> >
> > I thank the authors for their rebuttal, which contains important clarifications. While I do agree with other reviewers that the results are tied to the assumptions made, I believe the authors have provided reasonable justifications for their choices, and further experimental results would be required to truly validate the claims (and so I believe this paper could motivate experimentalists to pursue targeted multi-agent experiments). I maintain my score and overall positive opinion of the paper, and think it is a valuable contribution to NeurIPS.

---

> > > ### Author Response · Authors · 2024-08-12
> > >
> > > Thank you very much for your kind words, time, and consideration. We appreciate your encouraging words!

---

### Author Rebuttal · Authors · 2024-08-05

We thank the reviewers for their time and thoughtful comments. All reviewers found our work clear and novel, which we find encouraging. All reviewers identified similar weaknesses: this makes it clear that there are important ways our work can be strengthened. Here, we provide answers to the two major weaknesses shared by all reviewers. We additionally attach a pdf with figures that answer some specific questions from individual reviewers. A response to each individual reviewer’s comments is provided in the thread of the associated review.

**G1. Why summed place cell activations?**

The summing of place cell activations is certainly an important aspect of our model and one that we did not sufficiently motivate in our original submission. Mixed selectivity in the hippocampus has been observed in complex multi-agent settings [e.g. Forli and Yartsev, (2023)]. More generally, mixed selectivity is believed to play a broad role in neural computations [e.g. Rigotti et al. (2013)]. Therefore, we wanted a model with some mixing of place cell activity. We chose a sum (linear) for three reasons:

1. For the Sorscher et al. model to have grid cells, the place cell layer is required to have certain properties (e.g., zero spatial mean). Some non-linearities (e.g. sigmoid) that might be expected in the hippocampus - when multiple agents are in the environment - do not satisfy these properties. Therefore, to avoid trivially finding weakened grid cells, we chose summing the place cell activations. This is a simple way of meeting the restrictions, while still providing mixing.

2. Not having separate place cell populations forces the network to learn to integrate the correct velocities with the correct estimated positions of the agents. If instead, two disjoint populations of place cells are considered, then the RNN can achieve this by partitioning the units in half and developing two toroidal attractors. This is interesting, but not scalable. Additionally, partitioning the network would prevent the use of more flexible computations. For instance, an agent near the wall is restricted in its trajectory and therefore easier to predict. In this case, more units can be used to predict the trajectory of an agent in the middle of the environment, where more resolution may be helpful. However, this can happen only when the place cell population is shared across agents.

3. Since each place field’s width is relatively small (see Fig. 1C), many of the trajectories have place cell activations that are fully separate for the two agents. This is because the two agents are not close enough to activate the same place cells. Thus, for many trajectories, the summing of place cell activations does not deviate from the separability we might intuitively expect.

We will add these rationales into the main text (in Sec. 2) to help clarify and highlight this important choice.

**G2. Does MEC path integrate two agents simultaneously?**

It is entirely possible that MEC does not simultaneously path integrate two agents simultaneously. Until extensive neurophysiological and behavioral work is performed on this topic, it will remain an open question. However, there are several reasons why we think simultaneous dual agent path integration may occur in MEC:

1. Anecdotally, sports players often have to perform actions that require the integration of both their own motion and the motion of others. For instance, when a soccer/futbol player kicks the ball to a teammate, they have to understand both the direction and speed they are moving, as well as the direction and speed their teammate is moving. Similar examples can be seen when driving: switching lanes on the highway requires the maintenance of both where the driver’s car is going, as well as where nearby cars are going. Finally, recent experimental work by Alexander et al. (2022) showed that rats can learn to perform shortcuts (i.e., routes that are predictive and take less time) when trained to chase a laser pointer. These shortcuts require predicting where the laser pointer will go and where the self is going. Collectively, these observations suggest that the brain, in certain scenarios, performs path integration of multiple agents with little latency.

2. As noted in the main text, recent experimental work has found that MEC responds to the presence of other agents [Stangl et al., (2021); Wagner et al., (2023)]. In particular, Stangl et al. (2021) found that MEC had similar border responses whether an individual walked around an environment or watched another person walk around the environment. These results suggest that, if the brain is indeed able to path integrate multiple agents simultaneously, the MEC may be involved.

In our revised manuscript, we will clarify that simultaneous path integration is a hypothesis. We will include a section in the Appendix where we provide more details on why we think this hypothesis is reasonable, expanding upon points 1-2 above. In addition, we will emphasize in the Discussion section that the predictions of our dual agent RNN actually provide a way to test whether MEC does perform dual agent path integration. If the predictions are not supported by neurophysiological experiments, this could suggest that MEC does not perform dual agent path integration, or that it does so, but in a different reference frame – both of which are interesting alternatives.

**Additional note about Sorscher et al. RNN model**

As a last note, we mention that Nayebi et al. (2021) found that the Sorscher et al. RNN model outperformed many other models in explaining properties of recorded MEC activity. While the brain score metric used has its own limitations, this provides additional motivation for studying the Sorscher et al. RNN model, as it provides a better fit of MEC, beyond just grid cells. We will add this reference to the main text (Introduction) to further motivate the study of the Sorscher et al. model.

---

### Decision · Program_Chairs · 2024-09-25

**Decision:**

Accept (poster)

**Comment:**

This paper examines the spatial representations that emerge in RNNs trained to do "dual path integration", i.e. to perform path integration for two separate agents moving through space. The authors compare these dual path integrators to single-agent path integration models, and show that they develop distinct spatial representations. Specifically, the dual agents show less "classic" grid cell activity (both in terms of single neurons and population-level torus manifolds), and more relative encoding of spatial location for the two agents. These results have potential implications for the encoding of space in social contexts.

The reviewers raised some concerns about this paper, some of which were fairly consistent across reviewers. In particular, multiple reviewers asked why summed place cell activities should be used and why we should entertain the assumption that the entorhinal cortex may engage in path integration for others. The authors provided some additional reasoning on these questions, and also attended to other comments in the rebuttal. Following the rebuttal period, the final average score was 6, putting this paper in the accept range. The AC felt that the paper makes an interesting contribution, so given this and the final scores, a decision to accept was reached.